# Structural and functional properties of a magnesium transporter of the SLC11/NRAMP family

**Karthik Ramanadane, Monique S Straub, Raimund Dutzler\*, Cristina Manatschal\***

Department of Biochemistry, University of Zurich, Zurich, Switzerland

**Abstract** Members of the ubiquitous SLC11/NRAMP family catalyze the uptake of divalent transition metal ions into cells. They have evolved to efficiently select these trace elements from a large pool of $Ca^{2+}$ and $Mg^{2+}$, which are both orders of magnitude more abundant, and to concentrate them in the cytoplasm aided by the cotransport of $H^+$ serving as energy source. In the present study, we have characterized a member of a distant clade of the family found in prokaryotes, termed NRMTs, that were proposed to function as transporters of $Mg^{2+}$. The protein transports $Mg^{2+}$ and $Mn^{2+}$ but not $Ca^{2+}$ by a mechanism that is not coupled to $H^+$. Structures determined by cryo-EM and X-ray crystallography revealed a generally similar protein architecture compared to classical NRAMPs, with a restructured ion binding site whose increased volume provides suitable interactions with ions that likely have retained much of their hydration shell.

## Editor's evaluation

This work elegantly fuses cryo-EM, X-ray crystallography, and in vitro transport experiments to describe the structural basis for functional diversity in the SLC11/NRAMP family of membrane transporters. This work identifies factors responsible for selectivity of classical NRAMPS for transition metal ions (Fe, Mn) and the NRMT clade for alkali metal ion (Mg). Although selectivity is much discussed in transport of divalent metal ions, this is an outstanding example of a study that gets to the bottom of the structural determinants governing this behavior.

**\*For correspondence:**
dutzler@bioc.uzh.ch (RD);
c.manatschal@bioc.uzh.ch (CM)

**Competing interest:** The authors declare that no competing interests exist.

## Introduction

Divalent cations constitute important factors in numerous biological processes and they are thus essential nutrients that need to be imported into cells in the required amounts. From a chemical perspective, we distinguish the alkaline earth metals $Ca^{2+}$ and $Mg^{2+}$ from transition metal ions such as $Fe^{2+}$ and $Mn^{2+}$, which show distinct properties owing to their size and electronic structure. As major constituent of bone, $Ca^{2+}$ has to be taken up by vertebrates in high amount. Although its largest part is immobilized within the body, free $Ca^{2+}$ is present in the extracellular medium in low millimolar concentrations. In contrast, cytoplasmic $Ca^{2+}$ serves as second messenger and its distribution in the cell thus requires tight control (*Carafoli and Krebs, 2016*). $Mg^{2+}$ on the other hand is a ubiquitous ligand of nucleotides and nucleic acids (*de Baaij et al., 2015*; *Maguire and Cowan, 2002*). Despite the similar extracellular concentration, its abundance in the cytoplasm is several orders of magnitude higher compared to $Ca^{2+}$. In their interaction with other molecules, these alkaline earth metal ions are coordinated by electronegative hard ligands, preferentially oxygens, which compensate their positive charges by either full or partial countercharges (*Carafoli and Krebs, 2016*; *Payandeh et al., 2013*). In contrast to $Ca^{2+}$ and $Mg^{2+}$, which both constitute minor elements in the human body, transition metals are trace elements that are found at much

lower concentration, as they are required for specialized processes. Transition metals are distinguished by their incompletely filled d-orbitals. This property permits coordinative interactions with soft ligands containing free electron pairs and allows them to change their oxidation state, which makes them important cofactors in redox reactions. The transition metal ion $Fe^{2+}$ plays a central role in oxygen transport and, together with $Mn^{2+}$, it is a cofactor of enzymes catalyzing redox reactions. Due to their low abundance in the extracellular environment, the uptake of transition metals has thus to proceed with high selectivity to prevent competition with several orders of magnitude more abundant alkaline earth metals, which would prohibit their efficient accumulation. This challenge is overcome by specific transmembrane transport proteins, which catalyze the accumulation of transition metal ions inside cells. Among these proteins, members of the SLC11/NRAMP family play an important role. They are expressed in all kingdoms of life where they facilitate the transmembrane transport of different transition metal ions by a secondary active process that involves the cotransport of $H^+$ as energy source (*Courville et al., 2006*; *Nevo and Nelson, 2006*). In animals, these proteins are used for the transport of $Fe^{2+}$ (*Montalbetti et al., 2013*), whereas in prokaryotes the primary substrate is $Mn^{2+}$ (*Makui et al., 2000*). Besides the transport of both transition metals, plant NRAMPs are also involved in detoxification processes by transporting noxious metal ions such as $Cd^{2+}$ (*Huang et al., 2020a*).

The transport properties of SLC11/NRAMP proteins have been investigated in numerous functional studies (*Gunshin et al., 1997*; *Mackenzie et al., 2006*; *Makui et al., 2000*), supported by the structures of prokaryotic transporters determined in different conformations (*Bozzi et al., 2016b*; *Bozzi et al., 2019b*; *Ehrnstorfer et al., 2014*; *Ehrnstorfer et al., 2017*). With respect to its fold, the family is distantly related to other secondary active transporters that include the amino-acid transporter LeuT (*Ehrnstorfer et al., 2014*; *Yamashita et al., 2005*), although there is little relationship on a sequence level, SLC11/NRAMP structures provide detailed insight into substrate preference and transport mechanisms. Characterized family members show an exquisite selectivity for transition metals over alkaline earth metal ions, but they poorly discriminate between transition metals (*Ehrnstorfer et al., 2014*; *Gunshin et al., 1997*). This is accomplished by a conserved ion binding site that is located in the center of the protein (*Ehrnstorfer et al., 2014*). Several residues coordinate the largely dehydrated metal ion with hard, oxygen-containing ligands, which would equally well interact with other divalent cations. Additionally, the site allows interaction with the free electron pair of the thioether group of a conserved methionine residue, which serves as soft ligand that permits coordinative interactions with transition metal- but not alkaline earth metal ions (*Ehrnstorfer et al., 2014*). The mutation of this methionine to alanine in one of the bacterial homologues had a comparably small effect on transition metal ion transport but instead converted $Ca^{2+}$ into a transported substrate, which emphasizes the role of the residue in conferring ion selectivity (*Bozzi et al., 2016a*).

Although, throughout kingdoms of life, the majority of SLC11/NRMP transporters share similar functional properties, there are few family members that were described to transport with distinct substrate preference. Among these family members are prokaryotic proteins, termed NRAMP-related magnesium transporters (NRMTs), that were identified as uptake systems for $Mg^{2+}$. In the bacterium *Clostridium acetobutylicum*, the prototypic NRMT (CabNRMT) has permitted growth at limiting $Mg^{2+}$ concentrations, even in absence of any alternative transport pathways (*Shin et al., 2014*). The altered substrate preference is remarkable since, owing to its small ionic radius and the consequent high charge density, the interaction of $Mg^{2+}$ with its surrounding solvent is unusually strong, which makes the dehydration of the ion energetically costly and therefore requires unique features for its transport (*Chaudhari et al., 2020*; *Maguire and Cowan, 2002*). The uptake of $Mg^{2+}$ into cells is thus accomplished by few selective transport systems (*Payandeh et al., 2013*) that include the bacterial proteins CorA (*Lunin et al., 2006*), CorC (*Huang et al., 2021*), MgtE (*Hattori et al., 2007*) and their eukaryotic homologues (*Schäffers et al., 2018*; *Schweigel-Röntgen and Kolisek, 2014*), the TRP channel TRPM7 (*Huang et al., 2020b*) and $Mg^{2+}$-selective P-type pumps (*Maguire, 1992*). In the present study, we became interested in the mechanism of $Mg^{2+}$ transport by SLC11 proteins and we thus set out to investigate the functional and structural basis of this process by combining transport experiments with structural studies. We identified a biochemically well-behaved NRMT, which allowed us to characterize its transport behavior using in vitro transport experiments. Structure determination by cryo-electron microscopy and X-ray crystallography revealed an expanded substrate binding site that

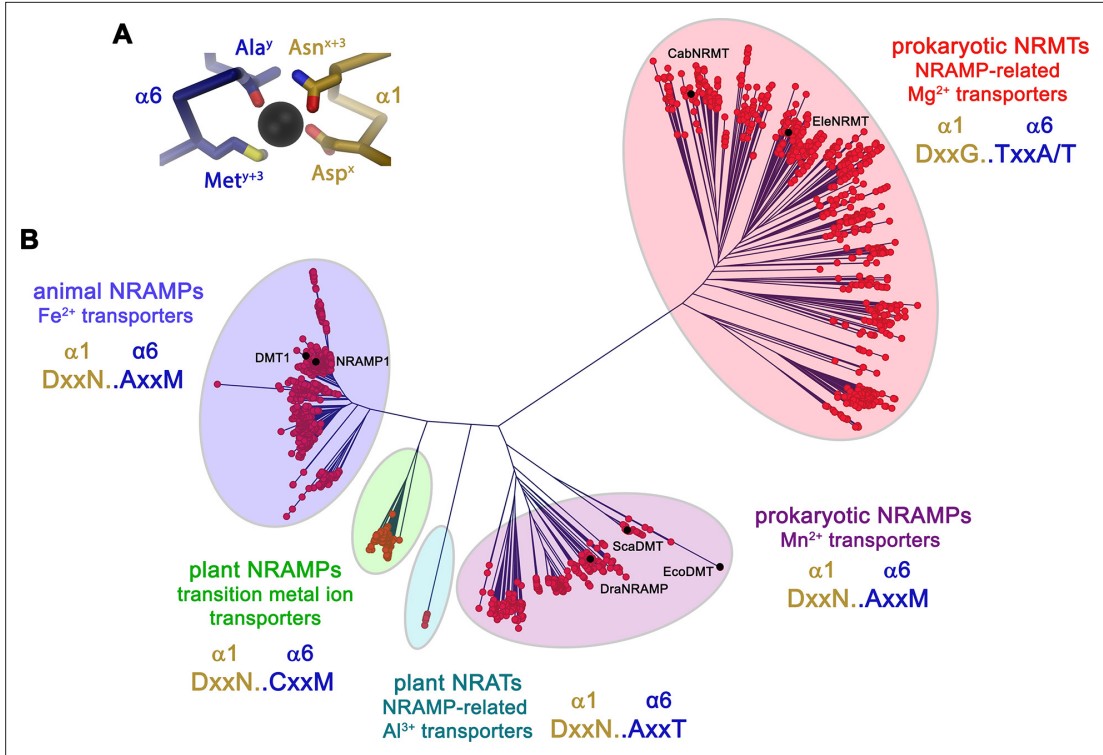

**Figure 1.** Phylogenetic analysis of the SLC11/NRAMP family. (**A**) Model of the consensus ion binding site of NRAMP transporters as obtained from the X-ray structure of ScaDMT (PDBID 5M95) in complex with Mn$^{2+}$. Regions of α1 and α6 are shown as Cα-trace, selected residues coordinating the ion as sticks. (**B**) Phylogenetic tree of SLC11 residues with different clades of the family highlighted. Selected family members are indicated. The consensus sequence of the respective ion binding site of each clade is shown.

The online version of this article includes the following figure supplement(s) for figure 1:

**Figure supplement 1.** Sequence alignment of selected protein regions of SLC11 family members.

binds the divalent cation as complex with most of its hydration shell, which permits selection against the larger Ca$^{2+}$, which is not a transported substrate.

## Results

### Phylogenetic analysis of NRAMP transporters

In light of the proposed altered substrate preference of certain prokaryotic SLC11 proteins, we became interested in the phylogeny of the family. For that purpose, we have identified numerous SLC11 homologues in BLAST searches of pro- and eukaryotic sequence databases using human DMT1, the prokaryotic Mn$^{2+}$ transporters EcoDMT and ScaDMT, the putative NRAMP-related aluminium transporter (NRAT) from *Orzya sativa* and the proposed Mg$^{2+}$ transporter CabNRMT as queries. In this way, sequences of 1100 eukaryotic and 447 prokaryotic NRAMP proteins and 745 prokaryotic NRMTs were aligned and subjected to phylogenetic characterization. The resulting phylogenetic tree displayed in *Figure 1* shows a distributed organization with different family members clustering into separated clades. Branches of this phylogenetic tree group proteins from the animal kingdom including human DMT1 and NRAMP1, which both function as H$^+$ coupled Fe$^{2+}$ transporters (*Forbes and Gros, 2003*; *Gunshin et al., 1997*), prokaryotic homologues including proteins of known structure such as ScaDMT (*Ehrnstorfer et al., 2014*), EcoDMT (*Ehrnstorfer et al., 2017*) and DraNRAMP (*Bozzi et al., 2019b*), whose primary function is H$^+$ coupled Mn$^{2+}$ transport and plant proteins ascribed to be involved in the transport of various transition metal ions such as the toxic metal Cd$^{2+}$ (*Huang et al., 2020a*). Within the plant kingdom, a small but separated branch encompasses putative Al$^{3+}$ transporters classified as NRATs (*Chauhan et al., 2021*; *Xia et al., 2010*). Finally, a large and distant clade of the family containing homologues

of the prototypic CabNRMT defines a group of transporters which, although clearly being part of the family, appear to have evolved to serve a different purpose (*Figure 1B*; *Shin et al., 2014*). The proposed functional distinction between different clades is manifested in the sequence of the substrate binding site that was identified in structural studies of prokaryotic SLC11 family members to coordinate divalent transition metals (*Ehrnstorfer et al., 2014*; *Figure 1A*, *Figure 1— figure supplement 1*). This site is defined by a signature containing a DxxN motif in the unwound center of α-helix 1 and an A/CxxM motif in the unwound center of α-helix 6. The signature is fully conserved in animal $Fe^{2+}$ transporters, bacterial $Mn^{2+}$ transporters and most transporters of plant origin, which all exhibit a strong selectivity against alkaline earth metal ions but poor discrimination between transition metal ions. Conversely, the motif is replaced by a DxxN-AxxT motif in NRATs and a DxxG-TxxA/T motif in NRMTs thus indicating that latter clades would exhibit a different substrate preference (*Figure 1—figure supplement 1*).

## Functional characterization of EcoDMT mutants

To gain further insight into the presumed substrate preference, we investigated how mutations of ion binding site residues would alter the selectivity of a classical NRAMP protein by characterizing the functional properties of the bacterial $Mn^{2+}$ transporter EcoDMT. To do so, we employed previously established proteoliposome-based transport assays with appropriate fluorophores to detect the uptake of specific ions into vesicles (*Ehrnstorfer et al., 2017*, *Figure 2—figure supplement 1*). All transport experiments were carried out in in presence of a 100-fold outwardly directed $K^+$ gradient, which after addition of the ionophore valinomycin establishes a membrane potential of –118 mV, to enhance the sensitivity of the applied assays.

Classical NRAMP proteins efficiently transport divalent transition metal ions with low micromolar affinity, while discriminating against alkaline earth metal ions such as $Ca^{2+}$ and $Mg^{2+}$ (*Ehrnstorfer et al., 2017*; *Mackenzie et al., 2006*). These functional properties are reflected in transport studies of EcoDMT, which employ the fluorophore calcein as efficient reporter to monitor micromolar concentrations of $Mn^{2+}$ whereas it is not responsive to either $Ca^{2+}$ or $Mg^{2+}$ even at much higher concentrations. In our experiments, we find a concentration-dependent quenching of the fluorescence due to $Mn^{2+}$ influx into vesicles containing the trapped fluorophore (*Figure 2A and B*, *Figure 2—figure supplement 1A*), which is neither inhibited by the addition of $Ca^{2+}$ nor $Mg^{2+}$ to the outside solution (*Figure 2C and D*). Latter would be expected if these ions would compete for the binding of $Mn^{2+}$. As suggested from previous studies (*Bozzi et al., 2016a*; *Ehrnstorfer et al., 2014*), we expected a key role of the methionine (Met 235) in the binding site of the protein as major determinant of this pronounced selectivity. We thus initially investigated whether it was possible to alter the substrate preference of EcoDMT by shortening the methionine side chain in the mutant EcoDMT M235A. This construct was still capable of transporting $Mn^{2+}$ with a threefold increase of its $K_m$ (*Figure 2A, B, E and G*), whereas now also $Ca^{2+}$ became a permeable substrate as illustrated by the competition of the ion with $Mn^{2+}$ transport (*Figure 2F and G*) and the direct detection of $Ca^{2+}$ transport using the fluorophore Fura2 (*Figure 2H*, *Figure 2—figure supplement 1B*). A similar result was previously obtained for the equivalent mutation of the homologue DraNRAMP (*Bozzi et al., 2016a*). In contrast to $Ca^{2+}$, we were not able to detect any interaction with $Mg^{2+}$ (*Figure 2I*), indicating that the energetic requirements to transport this smaller divalent cation are higher than for $Ca^{2+}$.

Finally, we attempted to confer $Mg^{2+}$ transport properties to EcoDMT by creating the triple mutant M235A/N54G/A232T, which converts its entire binding site to the residues found in prokaryotic NRMTs. As for the mutant M235A, this construct folded into a stable protein that retained its capability of $Mn^{2+}$ transport while remaining unable to interact with $Mg^{2+}$ (*Figure 2J–L*). In summary, the mutation of the binding site residues of EcoDMT towards an NRMT was well tolerated with the resulting constructs retaining their capability for $Mn^{2+}$ transport. This suggests that the structural and energetic requirements for interactions with this transition metal are lower than for alkaline earth metals. We also confirmed the role of the binding site methionine as an important determinant for the counterselection of $Ca^{2+}$, which due to its much higher concentration would outcompete $Mn^{2+}$ transport in a physiological environment. However, neither the replacement of the methionine nor the conversion of the binding site to residues observed in NRMTs was sufficient to facilitate $Mg^{2+}$ transport, demonstrating that structural features beyond the immediate binding site residues are required to transport this small divalent cation. Consequently, we decided to turn

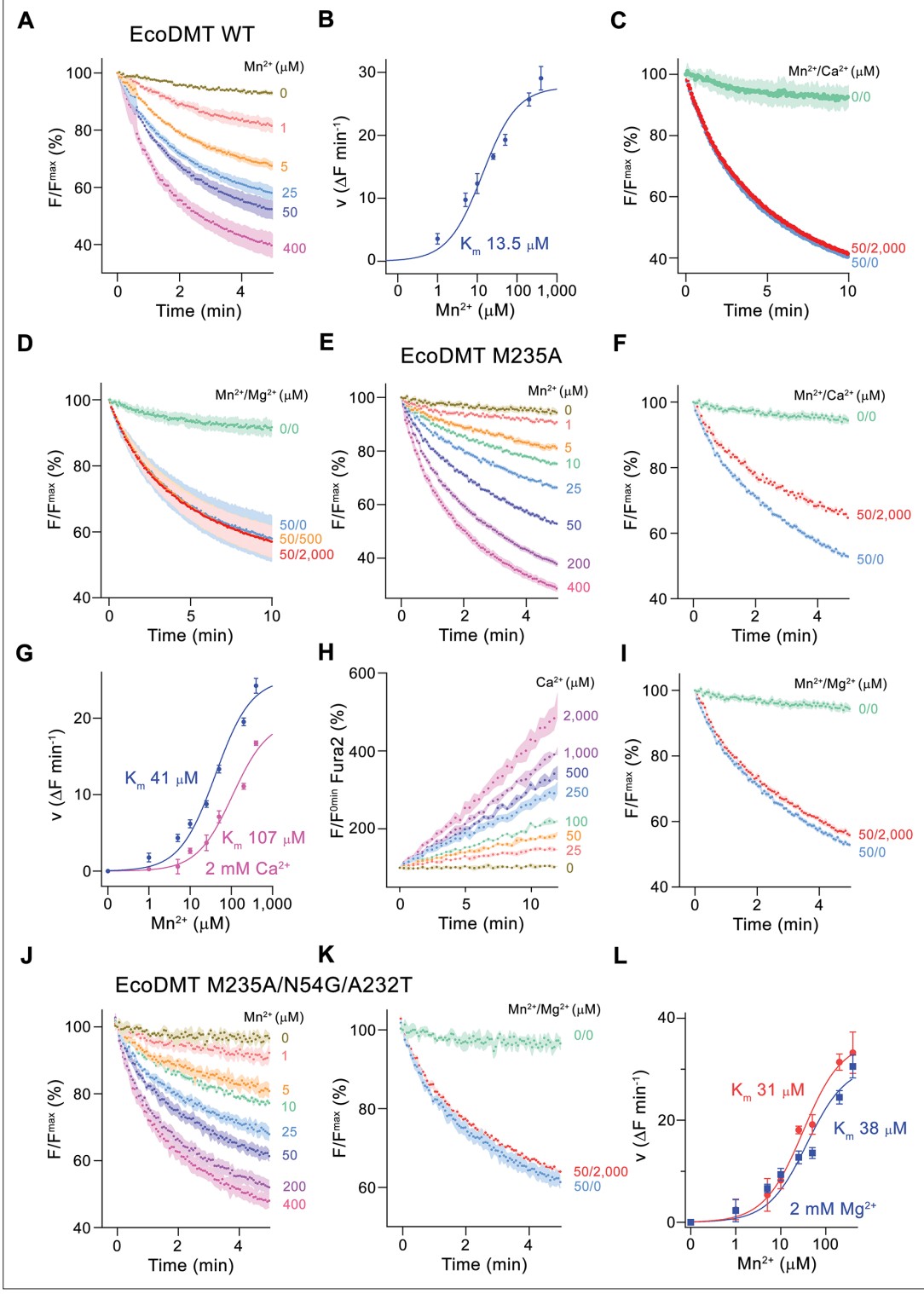

**Figure 2.** Functional characterization of EcoDMT mutants. (**A**) EcoDMT-mediated $Mn^{2+}$ transport into proteoliposomes. Data display mean of three experiments from three independent reconstitutions. (**B**) $Mn^{2+}$ concentration dependence of transport. Initial velocities were derived from individual traces of experiments displayed in (**A**), the solid line shows the fit to a Michaelis–Menten equation with an apparent $K_m$ of 13.5 µM. (**C**) $Mn^{2+}$ transport in presence of $Ca^{2+}$. Data display mean of seven experiments from two independent reconstitutions, except for the measurement without divalent ions (0/0, green, mean of two experiments). (**D**) $Mn^{2+}$ transport into EcoDMT proteoliposomes in presence of $Mg^{2+}$. Data display mean of 3 (0/0, green), 4 (50/0, blue)

*Figure 2 continued on next page*

*Figure 2 continued*

and nine experiments (50/500, orange and 50/2000, red) from three independent reconstitutions. (**E**) $Mn^{2+}$ transport into proteoliposomes containing the EcoDMT mutant M235A. Data display mean of eight experiments from three independent reconstitutions. (**F**) Inhibition of $Mn^{2+}$ transport in the mutant M235A by $Ca^{2+}$ (3 experiments from three independent reconstitutions). (**G**) $Mn^{2+}$ concentration dependence of transport into M235A proteoliposomes. For $Mn^{2+}$, initial velocities were derived from individual traces of experiments displayed in (**E**), for $Mn^{2+}$ in presence of 2 mM $Ca^{2+}$, data show mean of three experiments from three independent reconstitutions. The solid lines are fits to a Michaelis–Menten equation with apparent $K_m$ of 41 µM ($Mn^{2+}$) and 107 uM ($Mn^{2+}$ in presence of 2 mM $Ca^{2+}$). (**H**) $Ca^{2+}$-transport into M235A proteoliposomes assayed with the fluorophore Fura-2. Data display mean of tree experiments from three independent reconstitutions. (**I**) $Mn^{2+}$ transport in presence of $Mg^{2+}$. Data display mean of 6 experiments from three independent reconstitutions. (**J**) $Mn^{2+}$ transport into proteoliposomes containing the EcoDMT triple mutant M235A/N54G/A232T. Data display mean of three experiments from three independent reconstitutions. (**K**) Inhibition of $Mn^{2+}$ transport in the mutant M235A/N54G/A232T by $Mg^{2+}$ (six experiments from three independent reconstitutions). (**L**) $Mn^{2+}$ concentration dependence of transport into M235A/N54G/A232T proteoliposomes. For $Mn^{2+}$, initial velocities were derived from individual traces of experiments displayed in (**J**), for $Mn^{2+}$ in presence of 2 mM $Mg^{2+}$, data show mean of three (without $Mg^{2+}$, red) and six experiments (with $Mg^{2+}$, blue) from three independent reconstitutions. The solid lines are fits to a Michaelis–Menten equation with apparent $K_m$ of 31 µM ($Mn^{2+}$) and 38 µM ($Mn^{2+}$ in presence of 2 mM $Mg^{2+}$). (**A, C, D, E, F, I, J, K**) Uptake of $Mn^{2+}$ was assayed by the quenching of the fluorophore calcein trapped inside the vesicles. (**A, C, D, E, F, H, I, J, K**) Averaged traces are presented in unique colors. Fluorescence is normalized to the value after addition of substrate (t = 0). Applied ion concentrations are indicated. (**A–L**), Data show mean of the indicated number of experiments, errors are s.e.m.

The online version of this article includes the following figure supplement(s) for figure 2:

**Figure supplement 1.** Assay controls.

our attention towards the characterization of prokaryotic NRMTs to gain further insight into $Mg^{2+}$ transport by SLC11 homologues.

## Functional characterization of EleNRMT

Initially, our attempts toward the characterization of NRMTs were focused on the identification of a suitable candidate for structural and functional studies. Towards this end, we cloned and investigated the expression properties of 82 representative homologues selected from the pool of 745 sequences of putative NRMTs identified in BLAST searches (*Figure 1B*). Within this subset, we were able to single out a candidate from the bacterium *Eggerthella lenta* termed EleNRMT with promising biochemical properties (*Figure 3—figure supplement 1*). After expression and purification in the detergent n-dodecyl-beta-D-Maltoside (DDM), we found this protein to elute as monodisperse peak from a size exclusion chromatography column. (*Figure 3—figure supplement 2A*). Although much better behaved than the other investigated homologues, EleNRMT was still comparably instable in detergent solution and difficult to reconstitute into liposomes. We thus attempted to improve its stability by consensus mutagenesis (*Cirri et al., 2018*) by converting 11 positions identified in sequence alignments to their most abundant residues in NRMTs to create the construct EleNRMT^ts (*Figure 3—figure supplement 1A*). This approach improved the thermal stability of the monomeric protein and the efficiency of reconstitution without affecting its functional properties (*Figure 3*, *Figure 3—figure supplement 2B-D*). Due to the sensitivity of the assay and the fact that binding site mutants of EcoDMT retained their ability to transport $Mn^{2+}$, we initially investigated transport properties of EleNRMT for this divalent transition metal ion and found robust concentration-dependent activity that saturates with a $K_m$ of around 120 µM for either WT or the thermostabilized mutant (*Figure 3A, B, D and E*). For both constructs, we found $Mg^{2+}$ to inhibit $Mn^{2+}$ transport in a concentration-dependent manner, thus indicating that $Mg^{2+}$ would compete with $Mn^{2+}$ for its binding site (*Figure 3C and F*). We subsequently quantified binding of either ion to detergent solubilized protein by isothermal titration calorimetry and found a $K_d$ of around 100 µM for $Mn^{2+}$, which corresponds well with the measured $K_m$ of transport of the same ion and a somewhat lower affinity of around 400 µM for $Mg^{2+}$ (*Figure 3G and H*, *Figure 3—figure supplement 2E*, F). Together, transport and binding experiments demonstrate the capability of EleNRMT to interact with $Mg^{2+}$ thus providing evidence that it might indeed catalyze $Mg^{2+}$ uptake into bacteria.

To further characterize $Mg^{2+}$ transport by EleNRMT, we have quantified $Mn^{2+}$ uptake into proteoliposomes in presence of $Mg^{2+}$ and found a strong interference (*Figure 4A and B*). By directly assaying

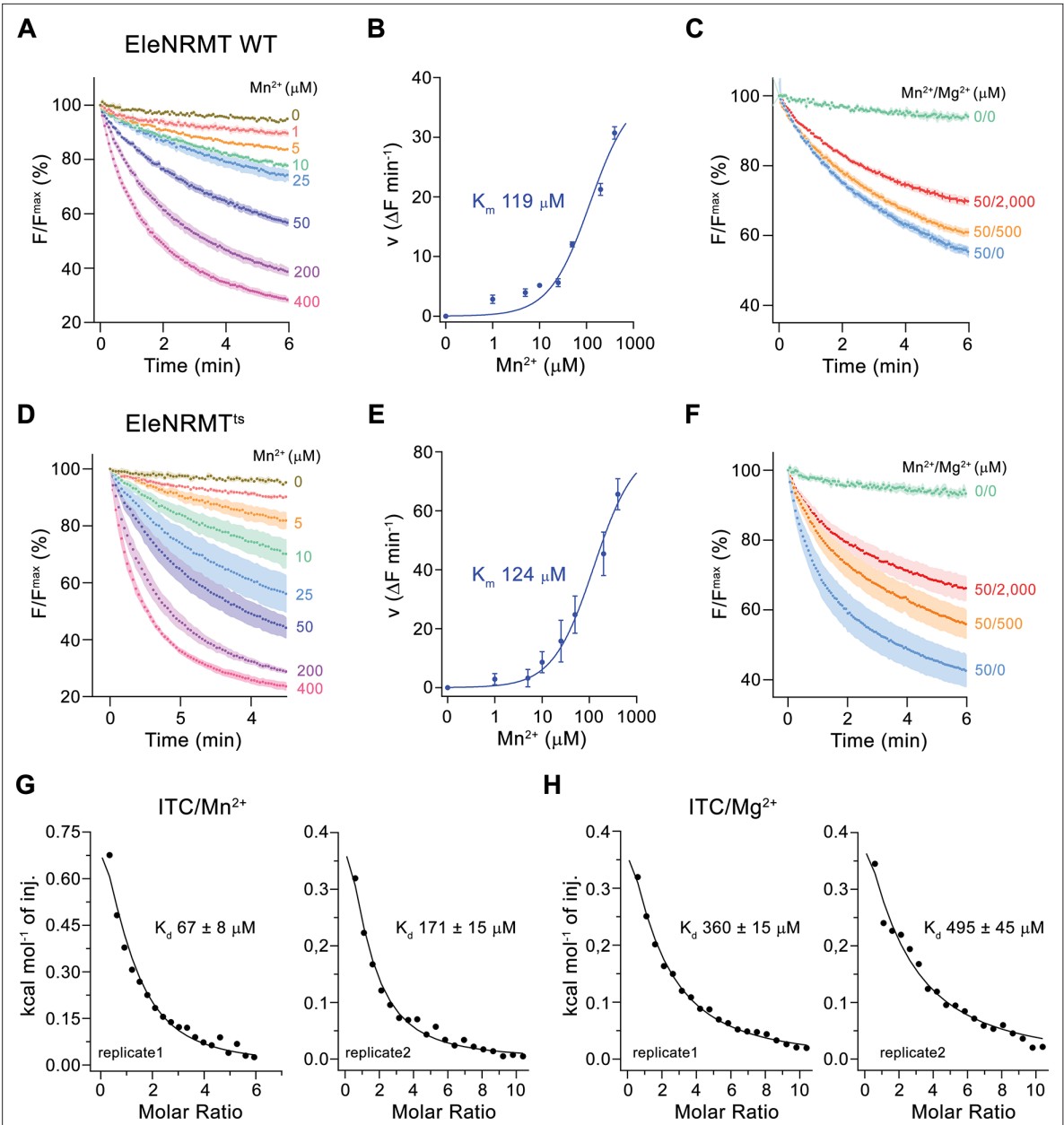

**Figure 3.** Transport properties of EleNRMT. (**A**) EleNRMT mediated $Mn^{2+}$ transport into proteoliposomes. Data display mean of three experiments from two independent reconstitutions. (**B**) $Mn^{2+}$ concentration dependence of transport. Initial velocities were derived from individual traces of experiments displayed in (**A**), the solid line shows the fit to a Michaelis–Menten equation with an apparent $K_m$ of 119 µM. (**C**) $Mn^{2+}$ transport in presence of $Mg^{2+}$. Data display mean of 3 (0/0, green), 6 (50/0, blue), 5 (50/500, orange), and 9 (50/2000, red) experiments from two independent reconstitutions. (**D**) $Mn^{2+}$ transport into proteoliposomes mediated by the thermostabilized mutant EleNRMT$^{ts}$. Data display mean of four experiments from two independent reconstitutions, except for the measurement with 25 µM $Mn^{2+}$ (mean of three experiments). (**E**) $Mn^{2+}$ concentration dependence of transport. Initial velocities were derived from individual traces of experiments displayed in (**D**), the solid line shows the fit to a Michaelis–Menten equation with an apparent $K_m$ of 124 µM. (**F**) $Mn^{2+}$ transport in presence of $Mg^{2+}$. Data display mean of 4 (0/0, green), 9 (50/0, blue), 11 (50/500, orange), and 8 (50/2000, red) experiments from two independent reconstitutions. A, C, D, F. Uptake of $Mn^{2+}$ was assayed by the quenching of the fluorophore calcein trapped inside the vesicles. Averaged traces are presented in unique colors. Fluorescence is normalized to the value after addition of substrate (t = 0). Applied ion concentrations are indicated. (**A–F**), Data show mean of the indicated number of experiments, errors are s.e.m. (**G–H**), Binding isotherms obtained from isothermal titrations of $Mn^{2+}$ (**G**) and $Mg^{2+}$ (**H**) to EleNRMT$^{ts}$. The data shown for two biological replicates per condition was fitted to a model assuming a single binding site with the binding isotherm depicted as solid line. Errors represent fitting errors.

The online version of this article includes the following figure supplement(s) for figure 3:

**Figure supplement 1.** EleNRMT sequence and topology.

**Figure supplement 2.** Stabilization and biochemical characterization of EleNRMT.

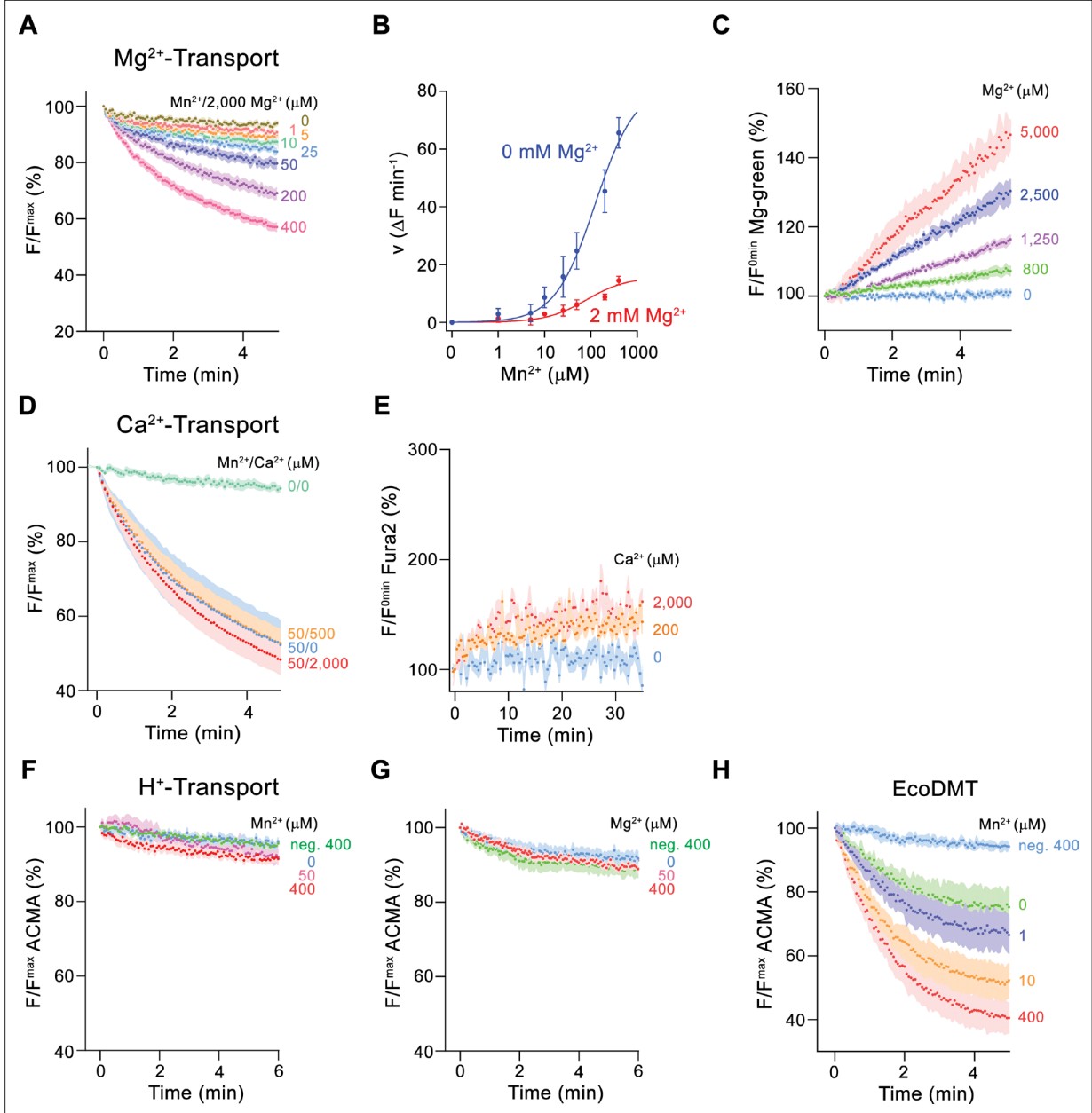

**Figure 4.** Ion selectivity and substrate coupling. (**A–E**) Metal ion transport by EleNRMT[ts]. (**A**) EleNRMT[ts] mediated $Mn^{2+}$ transport into proteoliposomes in presence of 2 mM $Mg^{2+}$. Data display mean of eight experiments from three independent reconstitutions, except for the measurement with 200 μM $Mn^{2+}$ (mean of seven experiments). (**B**) $Mn^{2+}$ concentration dependence of transport into EleNRMT[ts] proteoliposomes. For $Mn^{2+}$, data are as displayed in *Figure 3E*, for $Mn^{2+}$ in presence of 2 mM $Mg^{2+}$, initial velocities were derived from individual traces of experiments displayed in (**A**). The solid lines are fits to a Michaelis–Menten equation. (**C**) $Mg^{2+}$-transport into EleNRMT[ts] proteoliposomes assayed with the fluorophore Magnesium Green. Data display mean of 5 (0 μM $Mg^{2+}$), 8 (800 μM and 1250 μM $Mg^{2+}$), 4 (2500 μM $Mg^{2+}$), and 3 (5000 μM $Mg^{2+}$) experiments from two independent reconstitutions. (**D**) $Mn^{2+}$ transport in presence of $Ca^{2+}$. Data display mean of 4–11 experiments from three independent reconstitutions. (**A, D**) Uptake of $Mn^{2+}$ was assayed by the quenching of the fluorophore calcein trapped inside the vesicles. (**E**) $Ca^{2+}$-transport into EleNRMT[ts] proteoliposomes assayed with the fluorophore Fura-2. Data display mean of 3 (0 μM $Ca^{2+}$), 12 (200 μM $Ca^{2+}$) and 9 (2000 μM $Ca^{2+}$) experiments from 2 independent reconstitutions. The small increase of the fluorescence at high $Ca^{2+}$ concentrations likely results from non-specific leak into proteoliposomes. (**F–H**) Assay of H[+] transport with the fluorophore ACMA. Experiments probing metal ion coupled H[+] transport into proteoliposomes containing EleNRMT[ts] upon addition of $Mn^{2+}$ (eight experiments from two independent reconstitutions) (**F**) and $Mg^{2+}$ (eight experiments from two independent reconstitutions) (**H**) do not show any indication of H + transport. $Mn^{2+}$ coupled H[+] transport into EcoDMT proteoliposomes (three experiments from three independent reconstitutions and for the negative control [no protein, 400 μM $Mn^{2+}$], 4 measurements) is shown for comparison (**I**). A, C-H Averaged traces are presented in unique colors. Fluorescence is normalized to the value after addition of substrate (t = 0). Applied ion concentrations are indicated. (**A–H**), Data show mean of the indicated number of experiments, errors are s.e.m.

*Figure 4 continued on next page*

Figure 4 continued

The online version of this article includes the following figure supplement(s) for figure 4:

**Figure supplement 1.** Nanobody selection.

Mg²⁺ with the selective fluorophore Magnesium Green, we observed a concentration-dependent increase of the transport rate (**Figure 4C**, **Figure 2—figure supplement 1C**), with the lack of saturation at high substrate concentrations, reflecting the poor affinity of the dye for the ion. Since our experiments have confirmed Mg²⁺ as substrate of EleNRMT, we wondered whether the protein would also transport Ca²⁺ and thus investigated the inhibition of Mn²⁺ uptake by Ca²⁺ and the import of Ca²⁺ into proteoliposomes monitored with the fluorophore Fura-2 (**Figure 4D and E**). However, in contrast to Mg²⁺, we did in no case find strong evidence for the recognition of this larger alkaline earth metal ion, which has become a transported substrate for EcoDMT mutants lacking a conserved methionine in the binding site (**Figure 2F–H**). Latter finding is remarkable since the presumed metal ion binding site only consists of hard ligands that would also coordinate Ca²⁺ and it thus emphasizes the presence of distinct structural features of the pocket facilitating Mg²⁺ interactions (**Figure 3—figure supplement 1B**).

After confirming the capability of EleNRMT to specifically transport Mn²⁺ and Mg²⁺, we were interested whether the uptake of metal ions would be coupled to H⁺ as observed for previously characterized transition metal ion transporters of the family. To this end we employed the fluorophore ACMA to monitor pH changes that are induced by metal ion symport (**Figure 4F–H**). Whereas such concentration-dependent changes are readily observed for EcoDMT (**Figure 4H**), there was no acidification of proteoliposomes containing EleNRMT as a consequence of either Mg²⁺ or Mn²⁺ uptake (**Figure 4F and G**), thus suggesting that metal ion transport by EleNRMT is not coupled to H⁺. Collectively our functional experiments confirm the role of EleNRMT to function as uncoupled Mg²⁺ transporter. Due to the high-sequence conservation in this subclade of the SLC11 family, it is safe to assume that this property would likely also be shared by other NRMTs.

## Structural characterization of EleNRMT

Following the characterization of the transport properties of EleNRMT, we became interested in the molecular determinants underlying its distinct selectivity and thus engaged in structural studies by X-ray crystallography and cryo-EM. Since the monomeric protein, with a molecular weight of only 47 kDa and lacking pronounced domains extending from the membrane, is itself not a suitable target for cryo-EM, we initially attempted structural studies by X-ray crystallography but were unable to obtain crystals, despite extensive screening. To enlarge the size of the protein and facilitate crystallization, we generated specific nanobodies by immunization of alpacas with purified EleNRMT^ts followed by the assembly of a nanobody library from blood samples and selection by phage display. These experiments allowed us to identify three distinct binders recognizing the protein termed Nb1-3^EleNRMT (short Nb1-3) (**Figure 4—figure supplement 1A**). Despite the increase of the hydrophilic surface, which was frequently found to facilitate crystallization, all attempts to obtain crystals of EleNRMT^ts in complex with a single nanobody were unsuccessful. Similarly, efforts toward the structure determination of the same complexes by cryo-EM did not permit a reconstruction at high resolution. We thus attempted to further increase the size of the complex and found that Nb1 and either one of the two closely related nanobodies Nb2 or Nb3 were able to bind concomitantly to the protein to assemble into a ternary complex (**Figure 4—figure supplement 1B-D**). This complex showed improved crystallization properties and was of sufficient size to allow structure determination by single particle averaging. The structure of the tripartite EleNRMT^ts nanobody complex was determined by cryo-EM in absence of additional Mg²⁺ (EleNRMT^ts-Nb1,2) and in buffer containing 10 mM Mg²⁺ (EleNRMT^tsMg²⁺-Nb1,2). Datasets collected from respective samples allowed the 3D-reconstruction of a map at 3.5 Å for EleNRMT^ts-Nb1,2 and at 4.1 Å for EleNRMT^tsMg²⁺-Nb1,2 (**Figure 5—figure supplements 1 and 2**, **Table 1**). Both maps display equivalent conformations of the complex and permitted its unambiguous interpretation by an atomic model defining the features of an NRMT (**Figure 5A and B**, **Figure 5—figure supplement 3**). The structures show a protein complex consisting of the transporter EleNRMT^ts and two nanobodies binding to non-overlapping epitopes on the extracellular side (**Figure 5A and B**, **Figure 5—figure supplement 4**). This complex structure was subsequently used as search model for molecular

**Table 1.** Cryo-EM data collection, refinement and validation statistics.

| | Dataset 1<br>EleNRMT-NB<br>(EMDB-13985)<br>(PDB 7QIA) | Dataset 2<br>EleNRMT-NB Mg²⁺ complex<br>(EMDB-13987)<br>(PDB 7QIC) |
|---|---|---|
| **Data collection and processing** | | |
| Microscope | FEI Titan Krios | FEI Titan Krios |
| Camera | Gatan K3 GIF | Gatan K3 GIF |
| Magnification | 130'000 | 130'000 |
| Voltage (kV) | 300 | 300 |
| Electron exposure (e⁻/Å²) | 69.725 / 61 / 69.381 | 69.554 |
| Defocus range (µm) | –1.0 to –2.4 | –1.0 to –2.4 |
| Pixel size* (Å) | 0.651 (0.3255) | 0.651 (0.3255) |
| Initial number of micrographs (no.) | 22,117 | 12,427 |
| Initial particle images (no.) | 4 139 894 | 2 582 066 |
| Final particle images (no.) | 453 950 | 100 176 |
| Symmetry imposed | C1 | C1 |
| Map resolution (Å)<br>FSC threshold | 3.5<br>0.143 | 4.1<br>0.143 |
| Map resolution range (Å) | 2.9–7.0 | 3.3–8.3 |
| **Refinement** | | |
| Model resolution (Å)<br>FSC threshold | 3.5<br>0.5 | 4.1<br>0.5 |
| Map sharpening b-factor (Å²) | –184.4 | –159.2 |
| Model vs Map CC (mask) | 0.80 | 0.79 |
| Model composition<br>Non-hydrogen atoms<br>Protein residues<br>Ligand<br>Water | <br>4,843<br>639<br>0<br>3 | <br>4,841<br>639<br>1 (Mg)<br>0 |
| B factors (Å2)<br>Protein<br>Ligand<br>Water | <br>20.05<br>0<br>9.12 | <br>46.52<br>36.24<br>0 |
| R.m.s. deviations<br>Bond lengths (Å)<br>Bond angles (°) | <br>0.003<br>0.600 | <br>0.003<br>0.650 |
| Validation<br>MolProbity score<br>Clashscore<br>Poor rotamers (%) | <br>2.06<br>11.63<br>0 | <br>2.10<br>12.86<br>0.2 |
| Ramachandran plot<br>Favored (%)<br>Allowed (%)<br>Disallowed (%) | <br>92.10<br>7.90<br>0 | <br>92.26<br>7.74<br>0 |

*Values in parentheses indicate the pixel size in super-resolution.

replacement for a dataset collected from a crystal of the EleNRMT^ts-Nb1,2 complex grown in presence of 50 mM Mg²⁺ and extending to 4.1 Å (*Figure 5—figure supplement 5A*, *Table 2*). In the X-ray structure, both copies of the complex in the asymmetric unit closely resembled the cryo-EM model, thus providing an ideal system to determine the location of bound ions by anomalous

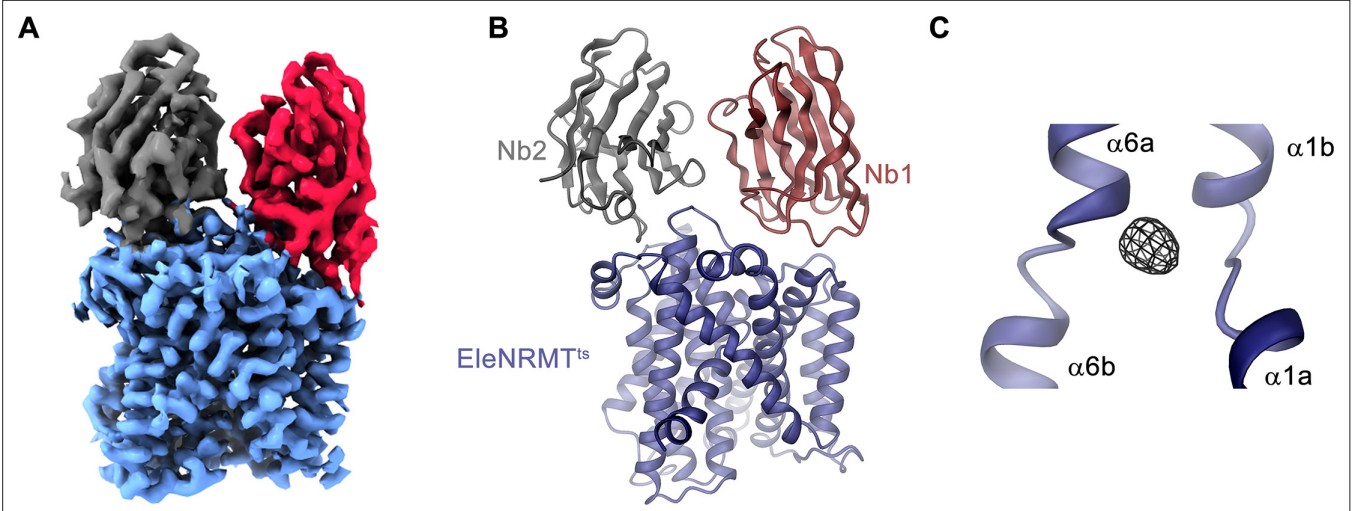

**Figure 5.** Structural characterization of EleNRMT[ts] by Cryo-EM and X-ray crystallography. (**A**) Cryo-EM density (contoured at 5 σ) of EleNRMT[ts] in complex with Nb1 and Nb2 at 3.5 Å viewed from within the membrane with the extracellular side on top and (**B**), ribbon representation of the complex in the same view. Proteins are shown in unique colors. (**C**) Anomalous difference density of $Mn^{2+}$ obtained from the crystal structure of the EleNRMT[ts]-Nb1-Nb2 complex (calculated at 5 Å and displayed as black mesh contoured at 4.5 σ). Model of EleNRMT[ts] encompassing the unwound center of helices α1 and α6 is shown as ribbon.

The online version of this article includes the following figure supplement(s) for figure 5:

**Figure supplement 1.** Cryo-EM reconstruction of the EleNRMT[ts]-nanobody complex.

**Figure supplement 2.** Cryo-EM reconstruction of EleNRMT[ts]-nanobody in presence of $Mg^{2+}$.

**Figure supplement 3.** Cryo-EM densities of EleNRMT-nanobody complexes.

**Figure supplement 4.** Nanobody binding interfaces.

**Figure supplement 5.** X-ray structures of EleNRMT-nanobody complexes in presence of $Mg^{2+}$ and $Mn^{2+}$ and comparison to cryo-EM structures.

scattering experiments (*Figure 5—figure supplement 5A-C*). Toward this end, we have soaked our crystals in solutions where $Mg^{2+}$ was replaced by $Mn^{2+}$ in the expectation that latter would occupy the binding site as suggested from functional experiments (*Figure 3*) and collected data at the anomalous absorption edge of the transition metal ion. This data allowed the unambiguous localization of $Mn^{2+}$ in the anomalous difference density, which showed a single strong peak per complex located in the equivalent position as previously identified for the binding site of ScaDMT (*Ehrnstorfer et al., 2014*, *Figure 5C*, *Figure 5—figure supplement 5B, C,*). In similar location, we find residual density in our cryo-EM datasets thus suggesting that $Mg^{2+}$ and $Mn^{2+}$ would both occupy the same site of the transporter (*Figure 5—figure supplement 5D, E*).

## EleNRMT structure and ion coordination

The structure of EleNRMT[ts] shows a transporter that displays the general organization of the SLC11 family. It encompasses 11 membrane-spanning segments, with the first 10 segments organized as two structurally related repeats of five α-helices that are inserted in the membrane with opposite orientations (*Figure 6A*, *Figure 3—figure supplement 1*). The protein adopts an inward-facing conformation that is illustrated in the comparison with the equivalent conformations of the transporters ScaDMT and DraNRAMP, which both superimpose with RMSDs of 2.2 and 2.0 Å, respectively (*Figure 6B*). This structure contains a large and continuous water-filled cavity that extends from the cytoplasm and leads to a pocket located in the center of the protein at the unwound parts of α-helices 1 and 6, which together form the ion binding site (*Figures 5C and 6C*). The location of the bound ion is defined in the anomalous difference density of $Mn^{2+}$ and its position presumably also corresponds to the binding position of $Mg^{2+}$ (*Figure 5C*, *Figure 5—figure supplement 5B-E*). Although the bound ion is found at an equivalent position as in the transporter ScaDMT, the binding pocket of EleNRMT is expanded as a consequence of the about 1.5 Å shift in the location of backbone atoms and differences in the volume of interacting side chains (*Figure 7A–C*). In both, $Mg^{2+}$ and $Mn^{2+}$ transporters, an aspartate on α-helix

**Table 2.** X-ray data collection and refinement statistics.

| | EleNRMT-Nb complex (PDB 7QJI) | EleNRMT-Nb complex Mn²⁺-soak (PDB 7QJJ) |
|---|---|---|
| **Data collection** | | |
| Space group | P21 | P21 |
| Cell dimensions | | |
| $a, b, c$ (Å) | 93.3, 122.2, 149.1 | 92.9, 115.9, 149.1 |
| $\alpha, \beta, \gamma$ (°) | 90, 107.8, 90 | 90, 107.6, 90 |
| Wavelength (Å) | 1.000 | 1.896 |
| Resolution (Å) | 50–4.1 (4.2–4.1) | 50–4.6 (4.7–4.6) |
| $R_{merge}$ | 12.5 (166.8) | 15.4 (147.6) |
| $I / \sigma I$ | 11.6 (2.4) | 10.8 (1.8) |
| Completeness (%) | 99.1 (98.9) | 99.8 (99.6) |
| Redundancy | 25.1 (27.1) | 14.0 (14.1) |
| **Refinement** | | |
| Resolution (Å) | 12–4.1 | 12–4.6 |
| No. reflections | 23,749 | 15,764 |
| $R_{work} / R_{free}$ | 26.7 / 31.5 | 28.4 / 32.8 |
| No. atoms | | |
| Protein | 9,632 | 9,632 |
| ligands | 0 | 2 |
| *B*-factors | | |
| Protein | 287.3 | 270.8 |
| ligands | - | 192.0 |
| R.m.s deviations | | |
| Bond lengths (Å) | 0.003 | 0.003 |
| Bond angles (°) | 0.60 | 0.60 |

Values in parentheses are for highest-resolution shell.

1 and a main chain interaction at the site of the unwound part of α-helix 6 remain a common feature of their binding sites, whereas the replacement of an alanine (Ala 223) in ScaDMT by a threonine (Thr 224) in EleNRMT introduces a side chain containing a hydroxyl group whose oxygen atom serves as additional ligand for ion interactions (*Figure 7B*). Further changes of other residues of the binding site alter its shape and volume. In ScaDMT, the bound ion is to a large extent excluded from the aqueous environment and instead tightly surrounded by residues of the binding site, whereas in EleNRMT the replacement of the asparagine side chain on α1 with a glycine and the methionine residue on α6 to alanine in conjunction with differences in the main chain positions widen the cavity to leave sufficient space for interacting water molecules (*Figure 7A–C*), suggesting that Mg²⁺ binds to the transporter in a partly solvated state retaining much of its first hydration shell. Thus, the EleNRMT structure implies that it is a combination of both, changes in the local environment conferred by the chemical nature of side chains and more delocalized changes causing an expansion of the binding site, which together are responsible for the altered selectivity.

Similar to the altered substrate preference, also the second functional hallmark of EleNRMT concerning the absent coupling of metal ion transport to protons, is manifested in its structure (*Figure 7D and E*). Proton transport in classical NRAMP transporters was proposed to be linked to structural features extending from the ion binding site (*Bozzi et al., 2019a*; *Bozzi et al., 2020*; *Bozzi*

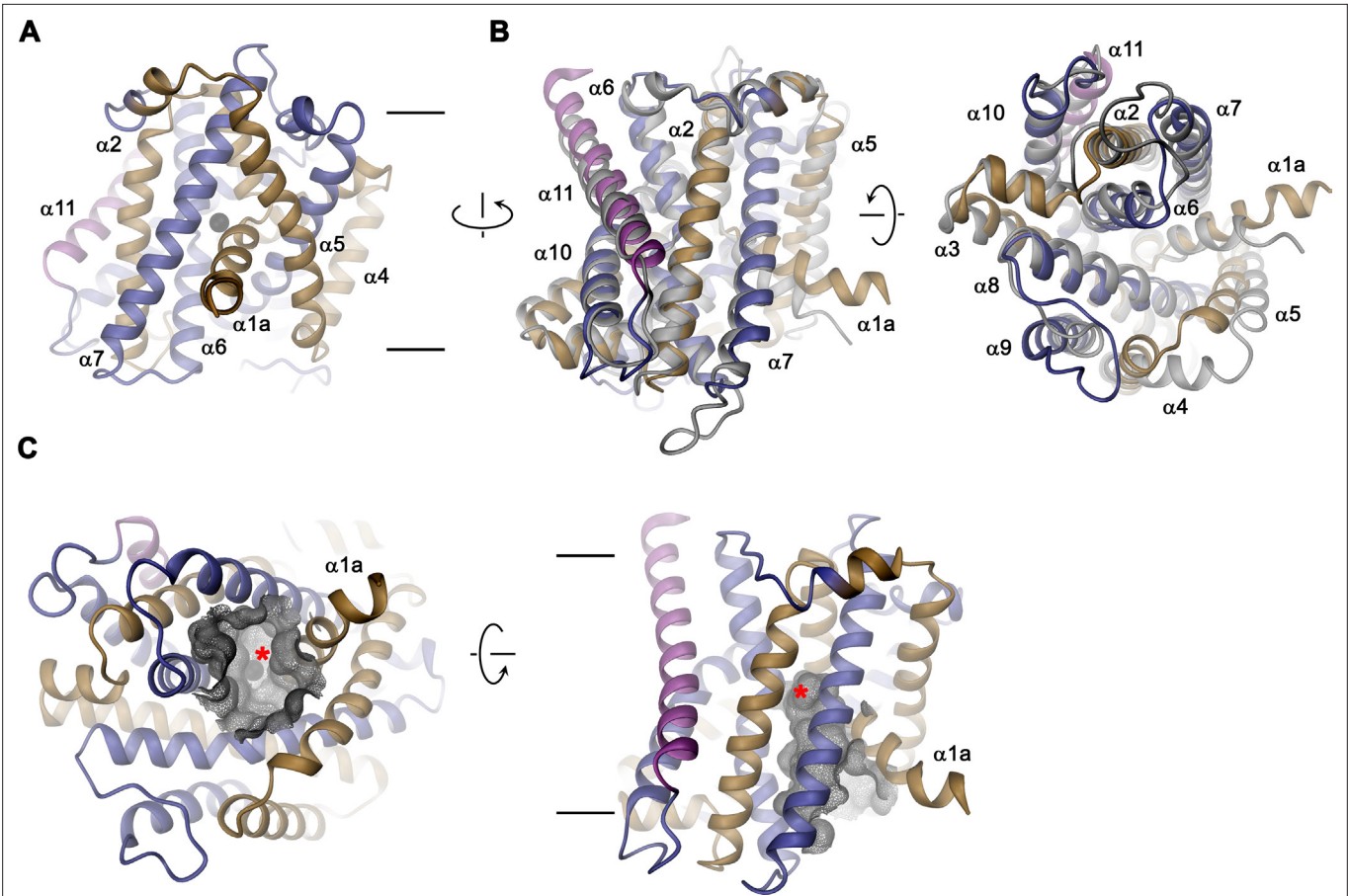

**Figure 6.** EleNRMT structure. (**A**) Ribbon representation of EleNRMT viewed from within the membrane with α1-α5 colored in beige, α6- α10 in blue and α11 in magenta. The position of a bound ion is indicated by a black sphere. (**B**) Superposition of EleNRMT and ScaDMT (based on PDBID:5M94 with modeled α1a), both showing equivalent inward-open conformations. Ribbon representation of ScaDMT is shown in gray, the representation of EleNRMT is as in A. (**C**) Ribbon representation of EleNRMT with surface of the aqueous cavity leading to the ion binding site displayed as mesh. Left, view from the cytoplasm and, right, from within the membrane. A bound ion is represented by a black sphere whose position is indicated with a red asterisk. A-C Selected helices are labeled. Membrane boundaries (**A, C**) and relationship between views (**A–C**) are indicated.

*et al., 2019b*; *Ehrnstorfer et al., 2017*; *Mackenzie et al., 2006*; *Pujol-Giménez et al., 2017*). These include the conserved aspartate involved in the coordination of metal ions, two conserved histidines on α-helix 6b downstream of the binding site methionine and a continuous path of acidic and basic residues on α-helices 3 and 9 surrounding a narrow aqueous cavity, which together appear to constitute an intracellular H$^+$ release pathway. Apart from the binding site aspartate (Asp 55) and glutamate on α-helix 3 (Glu 133) close to the metal ion binding site, all other positions in EleNRMT are altered to residues that cannot accept protons. The two histidines on α-helix 6b are substituted by a tryptophane and an asparagine, respectively, and the polar pathway residues on α-helix 3 and 9 are replaced by hydrophobic residues and a threonine (*Figure 7B and D*. *Figure 3—figure supplement 1B*).

## Functional properties of EleNRMT binding site mutants

After identifying the structural determinants of ion interactions in NRMTs, we were interested in the consequence of mutations of residues of the binding site on selectivity and thus investigated the transport properties of mutants. Consistent with experiments on other family members, the mutation of the aspartate in the binding site (Asp55), which is a universally conserved position in all SLC11 proteins, strongly compromises transport (*Figure 8A and B*). Similarly, the mutation of Gln 379 on α10, a conserved position that was proposed to be involved in ion coordination in the occluded state of DraNRAMP (*Bozzi et al., 2019b*), exerts a similarly strong effect (*Figure 8A and C*, *Figure 3—figure supplement 1B*), thus suggesting that EleNRMT undergoes an equivalent conformational change.

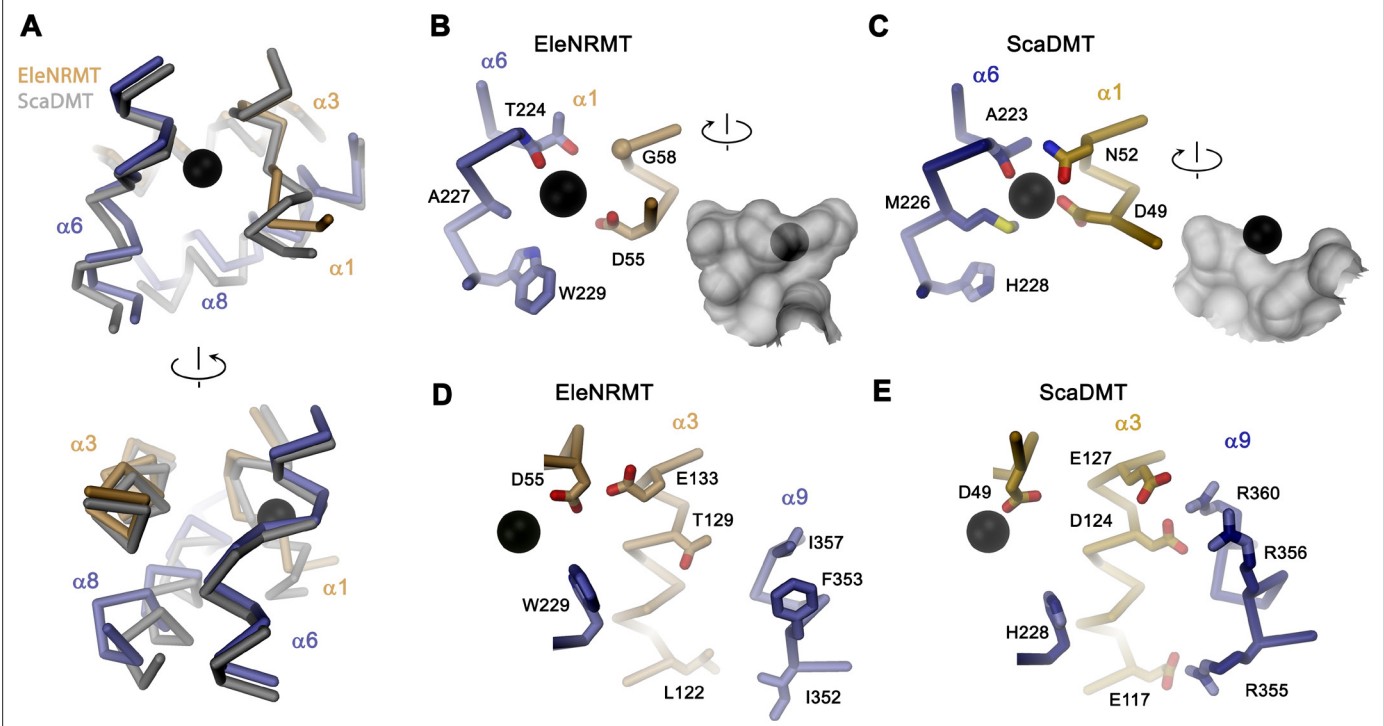

**Figure 7.** Comparison of regions of functional relevance between EleNRMT and ScaDMT. (**A**) Superposition of the metal ion binding regions of EleNRMT and ScaDMT (PDBID:5M95). The coloring is as in *Figure 6B*. The relationship between views is indicated. (**B**) Metal ion binding sites of EleNRMT (**B**) and ScaDMT (**C**). (**B, C**), inset (right) shows molecular surface surrounding the bound ion with indicated relationship. Whereas the ion in EleNRMT is located inside the aqueous cavity, its location in ScaDMT is outside of the cavity, tightly surrounded by coordinating residues. (**D**) Region of EleNRMT on α3 and α9 implicated in proton transport in NRAMPs and corresponding region in ScaDMT (**E**). (**D, E**) The bound ion with the conserved binding site aspartate on α1 and a residue corresponding to the first α6b histidine in NRAMPs, which was identified as potential H$^+$ acceptor, are shown for reference. (**A-E**) The proteins are depicted as Cα-trace, selected residues as sticks. The position of bound metal ions is represented as a black sphere. (**B-E**) Selected secondary structure elements and residues are indicated.

In contrast, the mutation of Thr 224 to alanine, the amino acid found in the equivalent position of NRAMP transporters, has smaller but still detectable impact on Mn$^{2+}$ transport, which is reflected in an at least two-fold increase of its $K_m$ (*Figure 8A, D and E*). In contrast to Mn$^{2+}$, this change has a much more pronounced effect on Mg$^{2+}$ interactions, which is manifested in the absence of appreciable inhibition of Mn$^{2+}$ transport in presence of increasing Mg$^{2+}$ concentrations and the non-detectable Mg$^{2+}$ transport when directly assaying this alkaline earth metal ion (*Figure 8F and G*). Finally, by replacing Gly 58 to asparagine in the background of the T224A mutant, we create an EleNRMT construct with a similar binding site signature as the EcoDMT mutant M235A, which showed increased Ca$^{2+}$ permeability. We thus investigated whether an equivalent conversion would also change the properties of EleNRMT with respect to its ability to transport Ca$^{2+}$, which is not a substrate of the WT protein (*Figure 4D and E*). In line with the pronounced effect already observed for T224A (*Figure 8D and E*), we see a further reduction of Mn$^{2+}$ transport in the double mutant T224A/G58N (*Figure 8H*). In contrast to the compromised Mn$^{2+}$ transport properties, this double mutant shows detectable Ca$^{2+}$ transport activity, thus illustrating that the conversion has introduced structural determinants that facilitate Ca$^{2+}$ interactions and that change the properties of the binding site to more closely resemble NRAMP proteins.

## Discussion

In our study, we have investigated the mechanistic basis of ion selectivity in the SLC11/NRAMP family. Whereas the bulk of the family efficiently transports divalent transition metal ions such as Fe$^{2+}$ or Mn$^{2+}$ from a high background of alkaline earth metal ions, there is an extended clade of the family in prokaryotes that has evolved as transport systems of the divalent cation Mg$^{2+}$ and that were thus

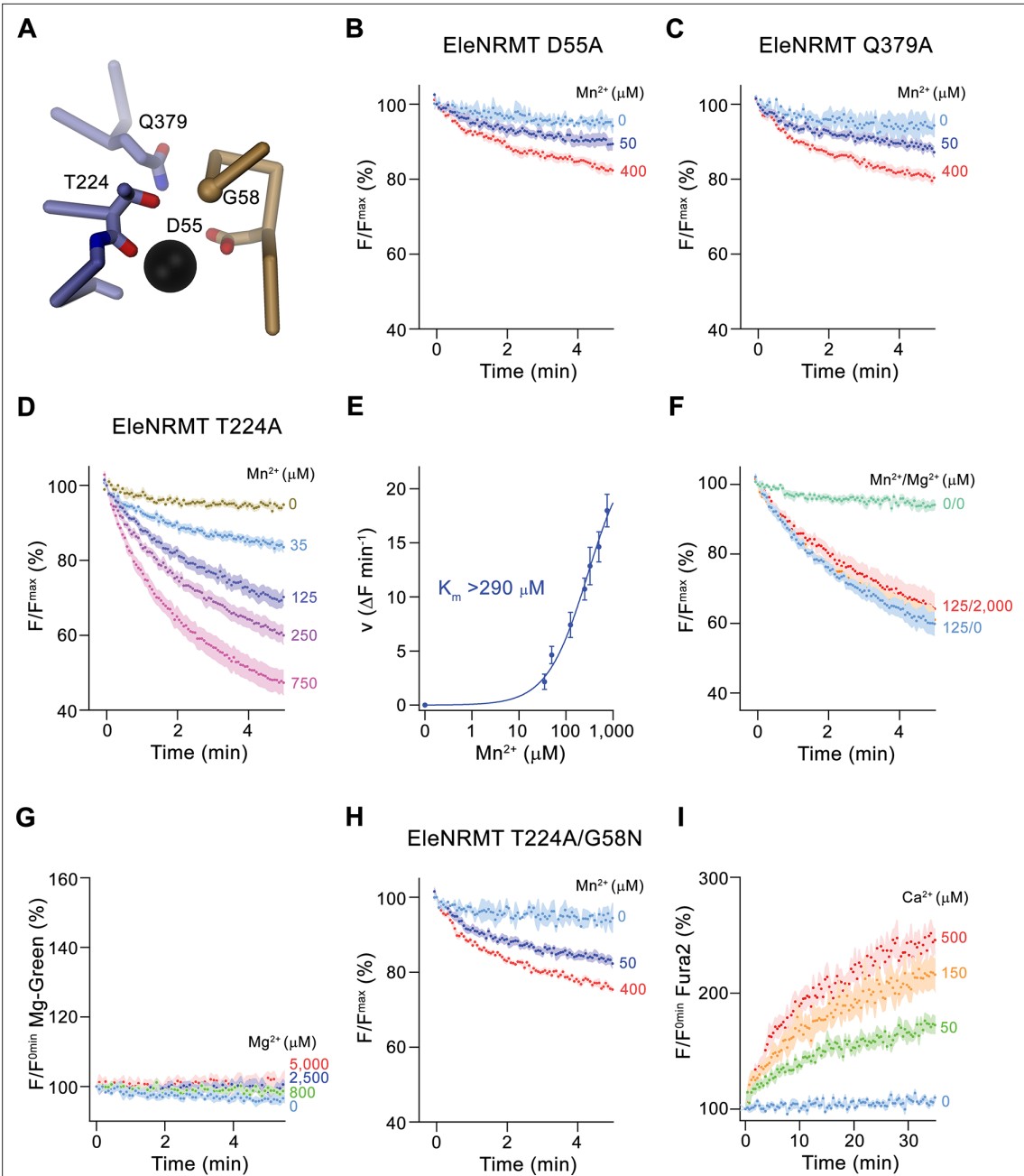

**Figure 8.** Functional characterization of binding site mutants. (**A**) Model of the ion binding site of EleNRMT with ion (black sphere) showing bound Mn$^{2+}$. Regions of α1 and α6 are shown as Cα-trace, selected residues coordinating the ion as sticks. (**B–D**) Mn$^{2+}$ transport mediated by EleNRMT$^{ts}$ binding site mutants. (**B**) D55A (2 (0 μM Mn$^{2+}$), 6 (50 μM Mn$^{2+}$), and 7 (400 μM Mn$^{2+}$) experiments from two independent reconstitutions), (**C**) Q379A (2 (0 μM Mn$^{2+}$), 6 (50 μM Mn$^{2+}$), and 7 (400 μM Mn$^{2+}$) experiments from two independent reconstitutions), (**D**) T224A (5 experiments from two independent reconstitutions). (**E**) Mn$^{2+}$ concentration dependence of transport of T224A. Initial velocities were derived from individual traces of experiments displayed in (**D**), the solid line shows the fit to a Michaelis–Menten equation with an apparent $K_m$ exceeding 290 μM. (**F**) Mn$^{2+}$ transport into T224A proteoliposomes in presence of Mg$^{2+}$ (4 experiments from two independent reconstitutions). (**G**) Mg$^{2+}$-transport into T224A proteoliposomes assayed with the fluorophore Magnesium green (4 experiments from two independent reconstitutions). (**H-I**) Transport mediated by the EleNRMT$^{ts}$ binding site mutant T224A/G58N. (**H**) Mn$^{2+}$ transport into proteoliposomes containing T224A/G58N (2 (0 μM Mn$^{2+}$), 6 (50 μM Mn$^{2+}$), and 7 (400 μM Mn$^{2+}$) experiments from two independent reconstitutions. (**I**) Ca$^{2+}$-transport into T224A/G58N proteoliposomes assayed with the fluorophore Fura-2 (9 (0 μM Ca$^{2+}$), 10 (50 μM Ca$^{2+}$), 11 (150 μM Ca$^{2+}$), and 10 (500 μM Ca$^{2+}$) experiments from two independent reconstitutions). (**B-D**), (**F- H**) Uptake of Mn$^{2+}$ was assayed by the quenching of the fluorophore calcein trapped inside the vesicles. (**B-D**), (**F-I**) Averaged traces are presented in unique colors. Fluorescence is normalized to value after addition of substrate (t = 0). Applied ion concentrations are indicated. (**B–I**), Data show mean of the indicated number of experiments, errors are s.e.m.

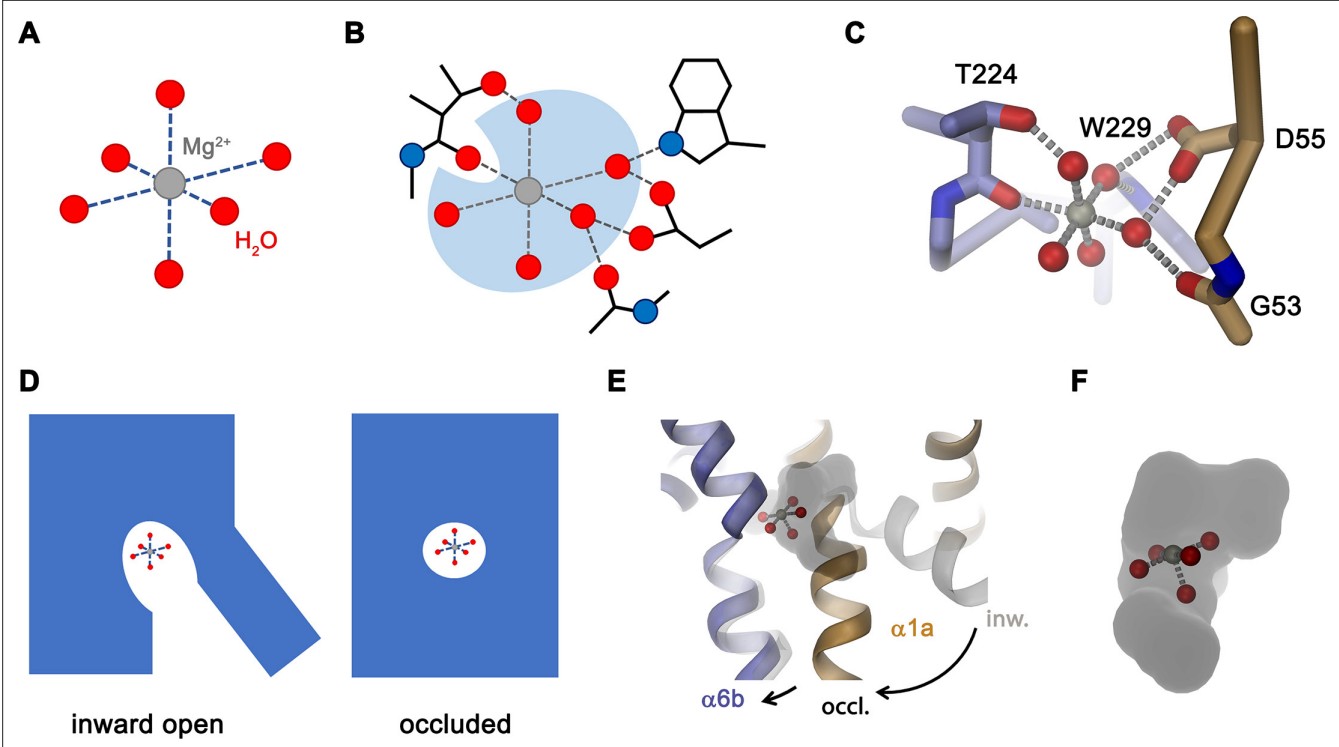

**Figure 9.** Features of Mg$^{2+}$ transport by NRMTs. Schematic depiction of (**A**), the octahedral coordination of the first hydration shell surrounding a Mg$^{2+}$ ion and (**B**), the same octahedral coordination within the binding site of a NRMT with five of the six water molecules remaining bound to the ion and where the bulk of protein interactions with the ion are mediated via coordinated water. (**C**) Model of Mg$^{2+}$ bound to the site of EleNRMT in the inward-facing conformation. The coordinates of the hydrated Mg$^{2+}$ were obtained from the high-resolution structure of the Mg$^{2+}$ transporter MgtE (PDBID 4U9L). The Mg$^{2+}$ ion was placed on the position of Mn$^{2+}$ obtained from the anomalous difference density defined in the X-ray structure. A single water molecule with a backbone oxygen atom was removed and the remaining complex was oriented to maximize interactions of coordinating waters with protein residues. Regions of the binding site are represented as Cα-trace and selected interacting residues as sticks. (**D**) Schematic depiction from an inward-facing to an occluded conformation of a NRMT with bound hydrated Mg$^{2+}$ indicated. (**E**) Hypothetical model of the transition of EleNRMT from an inward-facing (gray) to an occluded conformation (colored) that was based on the occluded conformation of DraNRAMP (PDBID: 6C3I). (**F**) Close-up of the binding pocket of the hypothetical model of the occluded conformation with bound Mg$^{2+}$–5H$_2$O complex.

termed NRAMP related Mg$^{2+}$ transporters (NRMTs) (*Shin et al., 2014*, *Figure 1*). These proteins are characterized by a distinct signature of residues constituting the ion binding site, located in the center of the protein in the unwound parts of the pseudo-symmetry related transmembrane helices α1 and α6 (*Figures 1 and 7*). We thus initially investigated whether we would be able to confer the capability to transport Mg$^{2+}$ to a transition metal ion transporter by mutation of its ion binding site.

The most striking difference in the ion binding site of both clades of the NRAMP family concerns the position of a conserved methionine that serves as a soft ligand for the interaction with transition metal ions, which is replaced by an alanine in NRMTs (*Figure 1*). The mutation of this residue in the classical NRAMP EcoDMT only moderately affected its capability to transport Mn$^{2+}$ and instead exerted its pronounced effect by converting Ca$^{2+}$ into a transported substrate (*Figure 2A–H*). This emphasizes the primary role of this residue to disfavor binding of this ubiquitous alkaline earth metal ion, which was already proposed in a previous study (*Bozzi et al., 2016a*). Similarly, an EcoDMT mutant where all three deviating residues of the binding site were mutated to their correspondents found in NRMTs was still efficient in Mn$^{2+}$ transport (*Figure 2E*), thus suggesting that, among divalent cations, the transport of transition metal ions might not depend on stringent structural prerequisites. In contrast to Ca$^{2+}$, neither of the mutants converted EcoDMT into a Mg$^{2+}$ transporter (*Figure 2D and K*), thus suggesting that the transport of latter might require specific features to cope with the unique chemical properties of this ion. Among divalent cations of biological relevance, Mg$^{2+}$ is distinguished by its small ionic radius, which results in an unusually strong interaction with the surrounding water molecules that would render its dehydration during transport energetically costly (*Persson, 2010*).

The first hydration shell of $Mg^{2+}$, which coordinates the ion in an octahedral geometry (*Figure 9A*), was thus suggested to remain bound to the ion in its initial encounter with the binding site and probably also during transport. A similar feature was indeed observed in the structures of the unrelated bacterial $Mg^{2+}$ transport proteins MgtE (*Takeda et al., 2014*) and CorA (*Guskov et al., 2012*) whereas protein interactions in the membrane-inserted domain of the transporter CorC are established with a fully dehydrated ion (*Huang et al., 2021*).

The distinct requirements for $Mg^{2+}$ transport are reflected in the properties of EleNRMT investigated in this study. Its capability to transport $Mn^{2+}$ (*Figure 3A–D and G*) underlines the described promiscuity of this ion in its interaction with proteins. In contrast, our studies have shown transport of $Mg^{2+}$ but not $Ca^{2+}$ (*Figures 3E, F, H , and 4A–E*) thus illustrating that both ions rely on distinct features of the protein which mediate their interaction. These are illustrated in the structure of EleNRMT in an inward-facing conformation determined by cryo-EM (*Figures 5 and 6*). In this structure, a wide aqueous cavity leads to an ion binding site whose spacious volume would preclude simultaneous interaction with a completely dehydrated ion (*Figures 6C, 7B and C*). The larger size of the site is an immediate consequence of the decreased side chain-volume of the surrounding residues concomitant with local differences in the backbone positions (*Figure 7A and B*), which explains why it was impossible to convert EcoDMT into a $Mg^{2+}$ transporter upon mutation of respective residues (*Figure 2D, I, K*). The position of a $Mn^{2+}$ ion bound to EleNRMT was defined by its anomalous difference density in a crystal structure of the protein (*Figure 5C*). Since $Mn^{2+}$ shows a similar octahedral coordination as $Mg^{2+}$, although with weaker affinity to the surrounding waters, we assume $Mg^{2+}$ to bind in an equivalent manner, which is also reflected in the presence of residual density in a dataset determined at high $Mg^{2+}$ concentration by cryo-EM (*Figure 5—figure supplement 5E*). Although the limited resolution of our structures does not permit assignment of water molecules, an octahedral $Mg^{2+}$-water complex model placed in the binding site suggests that only one of the solvent molecules would be substituted by a direct interaction with the oxygen of the amide backbone group of Thr 224, whereas other interactions with the conserved binding site would be mediated by coordinating waters (*Figure 9B and C*). The detailed structural properties of this cavity are expected to change upon a transition from an inward-open into an occluded intermediate conformation, which was observed in the corresponding structure of DraNRAMP, where the access of the binding site to the cytoplasm is closed (*Bozzi et al., 2019b*). However, in contrast to DraNRAMP, where the bound ion becomes tightly encircled by protein residues, the distinct structural features of EleNRMT and the lower side chain volume of residues of its binding site suggest that this cavity would be sufficiently big for $Mg^{2+}$ to retain much of its surrounding water during transport (*Figure 9D–F*) thus emphasizing that it is the larger hydrated complex of $Mg^{2+}$ rather than the naked ion that is relevant for transport.

A second pronounced feature that distinguishes EleNRMT and likely other NRMTs from classical SLC11 proteins concerns the absent coupling of metal ion transport to an energy source, which in case of NRAMPs is conferred by the cotransport of protons (*Ehrnstorfer et al., 2017*; *Gunshin et al., 1997*; *Figure 4F–H*). This property is reflected in the location of ionizable residues that are conserved within the proton-coupled part of the family and that are absent in EleNRMT (*Figure 7E and F*). It thus appears that in light of the high extracellular $Mg^{2+}$ concentration and the inside negative membrane potential such coupling may not be required for efficient uptake in the natural habitat of the bacterium.

Collectively, the presented study has provided detailed mechanistic insight into the structural basis of $Mg^{2+}$ transport in homologues of the SLC11/NRAMP family and it has confirmed previous results that have proposed this role based on biochemical considerations (*Shin et al., 2014*). Our work also illustrates the promiscuity of transition metal ion transport, which contrasts the tight requirements for the alkaline earth metal ions such as $Ca^{2+}$ and $Mg^{2+}$, latter of which is transported in a largely hydrated state. The observation that the previously investigated NRMT homologue did not permit growth upon $Mn^{2+}$ -limitation (*Shin et al., 2014*) could be a consequence of the lower affinity to this ion, the competition with abundant $Mg^{2+}$ and the absence of an energy source, which might be mandatory for efficient accumulation of the scarce substrate. Our study illustrates the specific requirements for the transport of $Mg^{2+}$ which are not met by classical NRAMP transporters. In this respect, it appears unlikely that the human transporter SLC11A1, which resides in the phagosomes of macrophages and

is involved in the immune response against invading microbes, would itself permit $Mg^{2+}$ transport as suggested in a recent study (**Cunrath and Bumann, 2019**).

# Materials and methods

## Key resources table

| Reagent type (species) or resource | Designation | Source or reference | Identifiers | Additional information |
|---|---|---|---|---|
| Chemical compound, drug | 1-palmitoyl-2-oleoyl-sn-glycero-3-phospho-(1'rac-glycerol) (18:1 06:0 POPG) | Avanti Polar Lipids | 840457 C | |
| Chemical compound, drug | *E. coli* polar extract | Avanti Polar Lipids | 100600 P | |
| Chemical compound, drug | 1-palmitoyl-2-oleoyl-sn-glycero-3-phosphoethanolamine (18:1 06:0 POPE) | Avanti Polar Lipids | 850757 C | |
| Chemical compound, drug | Triton X-100 | Sigma | Cat#T9284 | |
| Chemical compound, drug | Lysozyme | Applichem | Cat#A3711 | |
| Chemical compound, drug | Benzamidine | Sigma | B6506 | |
| Chemical compound, drug | Chloroform | Fluka | 25,690 | |
| Chemical compound, drug | DM | Anatrace | D322 | |
| Chemical compound, drug | DDM | Anatrace | D310S | |
| Chemical compound, drug | Diethyl ether | Sigma | 296,082 | |
| Chemical compound, drug | DNase I | AppliChem | A3778 | |
| Chemical compound, drug | glycerol 99% | Sigma | G7757 | |
| Chemical compound, drug | HCl | Merck Millipore | 1.00319.1000 | |
| Chemical compound, drug | HEPES | Sigma | H3375 | |
| Chemical compound, drug | Imidazole | Roth | X998.4 | |
| Chemical compound, drug | L-(+)-arabinose | Sigma | A3256 | |
| Chemical compound, drug | Leupeptin | AppliChem | A2183 | |
| Chemical compound, drug | Pepstatin | AppliChem | A2205 | |
| Chemical compound, drug | Valinomycin | Thermofischer Scientific | V1644 | |
| Chemical compound, drug | Calcimycin | Thermofischer Scientific | A1493 | |
| Chemical compound, drug | Ionomycin | Thermofischer Scientific | I24222 | |
| Chemical compound, drug | Fura2 | Thermofischer Scientific | F1200 | |
| Chemical compound, drug | Magnesium Green | Thermofischer Scientific | M3733 | |
| Chemical compound, drug | ACMA | Thermofischer Scientific | A1324 | |
| Chemical compound, drug | CCCP | Sigma | C2759 | |
| Chemical compound, drug | Calcein | Thermofischer Scientific | C481 | |
| Chemical compound, drug | Phosphate buffered saline | Sigma | D8537 | |
| Chemical compound, drug | PEG400 | Sigma | 91,893 | |
| Chemical compound, drug | Manganese acetate | Sigma | 330,825 | |
| Chemical compound, drug | Magnesium acetate | Fluka | 63,047 | |
| Chemical compound, drug | Calcium Chloride | Sigma | 223,506 | |
| Chemical compound, drug | Manganese Chloride | Fluka | 31,422 | |
| Chemical compound, drug | Magnesium Chloride | Fluka | 63,065 | |
| Chemical compound, drug | PMSF | Sigma | P7626 | |
| Chemical compound, drug | Potassium chloride | Sigma | 746,346 | |
| Chemical compound, drug | Sodium chloride | Sigma | 71,380 | |
| Chemical compound, drug | Terrific broth | Sigma | T9179 | |

*Continued on next page*

*Continued*

| Reagent type (species) or resource | Designation | Source or reference | Identifiers | Additional information |
|---|---|---|---|---|
| Commercial assay or kit | 4%–20% Mini-PROTEAN TGX Precast Protein Gels, 15-well, 15 µl | BioRad Laboratories | 4561096DC | |
| Commercial assay or kit | Amicon Ultra-4 Centrifugal Filters Ultracel 10 K, 4 ml | Merck Millipore | UFC801096 | |
| Commercial assay or kit | Amicon Ultra-4 Centrifugal Filters Ultracel 50 K, 4 ml | Merck Millipore | UFC805096 | |
| Commercial assay or kit | Amicon Ultra-4 Centrifugal Filters Ultracel 100 K, 4 ml | Merck Millipore | UFC810024 | |
| Commercial assay or kit | Biobeads SM-2 adsorbents | BioRad Laboratories | 152–3920 | |
| Commercial assay or kit | Avestin Extruder kit | Sigma | Cat#Z373400 | |
| Commercial assay or kit | 400 nm polycarbonate filters | Sigma | Cat#Z373435 | |
| Commercial assay or kit | 96-well black walled microplates | Thermofischer Scientific | Cat#M33089 | |
| Commercial assay or kit | Ni-NTA resin | ABT Agarose Bead Technologies | 6BCL-NTANi-X | |
| Commercial assay or kit | QuantiFoil R1.2/1.3 Au 200 mesh | Electron Microscopy Sciences | Q2100AR1.3 | |
| Commercial assay or kit | Superdex 200 10/300 GL | Cytiva | 17517501 | |
| Commercial assay or kit | Superdex 200 Increase 3.2/300 | Cytiva | 28990946 | |
| Commercial assay or kit | Superdex 200 Increase 5/150 GL | Cytiva | 28990945 | |
| Commercial assay or kit | Superdex 75 10/300 GL | Cytiva | 17517401 | |
| Other | BioQuantum Energy Filter | Gatan | NA | |
| Other | HPL6 | Maximator | NA | |
| Other | K3 Direct Detector | Gatan | NA | |
| Other | Titan Krios G3i | ThermoFisher Scientific | NA | |
| Other | Viber Fusion FX7 imaging system | Witec | NA | |
| Other | TECAN M1000 Infinite | TECAN | NA | |
| Other | Vitrobot Mark IV | ThermoFisher Scientific | NA | |
| Other | µDAWN MALS Detector | Wyatt Technology | NA | |
| Recombinant DNA reagent | gDNA Eggerthella lenta | DSMZ | 2,243 | |
| Recombinant DNA reagent | gDNA Eremococcus coleocola | DSMZ | 15,696 | |
| Recombinant DNA reagent | Bacterial expression vector with C-terminal 3 C cleavage site, GFP-tag and His-tag | Dutzler laboratory | NA | |
| Recombinant DNA reagent | Bacterial expression vector with N-terminal His-tag and 3 C cleavage site | Dutzler laboratory | NA | |
| Recombinant DNA reagent | Bacterial expression vector with N terminal pelB sequence, His-tag, MBP, 3 C cleavage site | Dutzler laboratory | NA | |
| Recombinant protein | HRV 3 C protease | Expressed (pET_3 C) and purified in Dutzler laboratory | NA | |
| Software, algorithm | 3DFSC | *Tan et al., 2017* | https://3dfsc.salk.edu/ | |
| Software, algorithm | ASTRA7.2 | Wyatt Technology | https://www.wyatt.com/products/software/astra.html | RRID:SCR_016255 |
| Software, algorithm | Chimera v.1.15 | *Pettersen et al., 2004* | https://www.cgl.ucsf.edu/chimera/ | RRID:SCR_004097 |
| Software, algorithm | ChimeraX v.1.1.1 | *Pettersen et al., 2021* | https://www.rbvi.ucsf.edu/chimerax/ | RRID:SCR_015872 |
| Software, algorithm | Coot v.0.9.4 | *Emsley et al., 2010* | https://www2.mrc-lmb.cam.ac.uk/personal/pemsley/coot/ | RRID:SCR_014222 |
| Software, algorithm | cryoSPARC v.3.0.1/v.3.2.0 | Structural Biotechnology Inc. | https://cryosparc.com/ | RRID:SCR_016501 |
| Software, algorithm | DINO | | http://www.dino3d.org | RRID:SCR_013497 |
| Software, algorithm | EPU2.9 | ThermoFisher Scientific | NA | |
| Software, algorithm | Phenix | *Liebschner et al., 2019* | https://www.phenix-online.org/ | RRID:SCR_014224 |
| Software, algorithm | RELION 3.0.7 | *Zivanov et al., 2018* | https://www3.mrc-lmb.cam.ac.uk/relion/ | RRID:SCR_016274 |

*Continued on next page*

*Continued*

| Reagent type (species) or resource | Designation | Source or reference | Identifiers | Additional information |
|---|---|---|---|---|
| Software, algorithm | JALVIEW | *Waterhouse et al., 2009* | | |
| Software, algorithm | Muscle | *Edgar, 2004* | | |
| Strain, strain background (E coli) | *E. coli* MC1061 | ThermoFisher Scientific | C66303 | |

## Phylogenetic analysis of the SLC11 family

The protein sequences of ScaDMT (UniprotKB A0A4U9TNH6), EcoDMT (UniprotKB E4KPW4), hDMT1 (UniprotKB P49281), OsNRAT1 (Uniprot KB Q6ZG85), CabNRMT-c0685 (UniprotKB Q97L77), and CabNRMT-c3329 (UniprotKB Q97DZ1) were respectively used as query sequences in BLASTp searches in diverse sequence databases. In that manner, 1100 eukaryotic and 447 prokaryotic standard SLC11 transporters and 745 prokaryotic NRMTs were selected and aligned with CLC Main Workbench (QIAGEN). The identity of the 745 prokaryotic NRMTs was confirmed by individual inspection of the proposed ion binding site. A multiple sequence alignment was used to generate a phylogenetic tree using the neighbor joining cluster generation method. Protein distances were measured using a Jukes-Cantor model and a Bootstrapping analysis was performed with 100 replicates.

## Expression and purification of EcoDMT

EcoDMT and its mutants were expressed and purified as described previously (*Ehrnstorfer et al., 2017*). Briefly, the vector EcoDMT-pBXC3GH (Invitrogen) was transformed into *E. coli* MC1061 cells and a preculture was grown in TB-Gly medium overnight using ampicillin (100 µg/ml) as a selection marker for all expression cultures. The preculture was used at a 1:100 volume ratio for inoculation of TB-Gly medium. Cells were grown at 37 °C to an $OD_{600}$ of 0.7–0.9, after which the temperature was reduced to 25 °C. Protein expression was induced by addition of arabinose to a final concentration of 0.004% (w/v) for 12–14 hr at 18 °C. Cells were harvested by centrifugation for 20 min at 5,000 g, resuspended in buffer RES (200 mM NaCl, 50 mM KPi pH 7.5, 2 mM $MgCl_2$, 40 µg/ml DNAseI, 10 µg/ml lysozyme, 1 µM leupeptin, 1 µM pepstatin, 0.3 µM aprotinin, 1 mM benzamidine, 1 mM PMSF) and lysed using a high pressure cell lyser (Maximator HPL6). After a low-speed spin (10,000 g for 20 min), the supernatant was subjected to a second centrifugation step (200,000 g for 45 min). The pelleted membrane was resuspended in buffer EXT (200 mM NaCl, 50 mM KPi pH 7.5, 10% glycerol) at a concentration of 0.5 g of vesicles per ml of buffer. The purification of EcoDMT and its mutants was carried out at 4 °C. Isolated membrane fractions were diluted 1:2 in buffer EXT supplemented with protease inhibitors (Roche cOmplete EDTA-free) and 1% (w/v) n-decyl-β-D-maltopyranoside (DM, Anatrace). The extraction was carried out under gentle agitation for 2 hr at 4 °C. The lysate was cleared by centrifugation at 200,000 g for 30 min. The supernatant supplemented with 15 mM imidazole at pH 7.5 was subsequently loaded onto NiNTA resin and incubated for at least 1 hr under gentle agitation. The resin was washed with 20 column volumes (CV) of buffer W (200 mM NaCl, 20 mM HEPES pH 7, 10% Glycerol, 50 mM imidazole pH 7.5, 0.25% (w/v) DM) and the protein was eluted with buffer ELU (200 mM NaCl, 20 mM HEPES pH 7, 10% Glycerol, 200 mM imidazole pH 7.5, 0.25% (w/v) DM). Tag cleavage proceeded by addition of HRV 3 C protease at a 3:1 molar ratio and dialysis against buffer DIA (200 mM NaCl, 20 mM HEPES pH7, 10% Glycerol, 0.25% (w/v) DM) for at least 2 hr at 4 °C. After application to NiNTA resin to remove the tag, the sample was concentrated by centrifugation using a 50 kDa molecular weight cut-off concentrator (Amicon) and further purified by size exclusion chromatography using a Superdex S200 column (GE Healthcare) pre-equilibrated with buffer SEC (200 mM NaCl, 20 mM HEPES pH 7, 0.25% (w/v) DM). The peak fractions were pooled, concentrated, and used directly without freezing.

## Expression screening of NRMT homologues

The DNA coding for NRMTs from 82 bacterial and archaeal strains were cloned from gDNA (obtained from DSMZ) into pBXNHG3 or pBXC3H vectors (Invitrogen) using FX cloning (*Geertsma and Dutzler, 2011*) providing a final construct where an NRMT contains a 10-His tag, a GFP and a HRV 3 C protease cleavage site fused to its N-terminus (pBXNHG3) or a HRV 3 C protease cleavage site and a 10-His tag fused to its C-terminus (pBXC3H). Both vectors contain an arabinose promoter and an ampicillin

resistance gene. Consequently, all cultures were supplemented with 100 µg/ml ampicillin as selection marker. The vectors were transformed into *E. coli* MC1061 cells and precultures were grown in Terrific Broth (TB) medium overnight. The small-scale expression was performed in 24-well plates with gas permeable adhesive covers in TB medium supplemented with 0.5% glycerol (TB-Gly). Each culture was inoculated at a 1:100 volume ratio. The cells were grown at 37 °C to an $OD_{600}$ of 0.7–0.9, before the temperature was reduced to 25 °C. The expression was induced by addition of arabinose to a final concentration of 0.001% (w/v) for 12–14 hr at 18 °C. Cells were harvested by centrifugation for 20 min at 5,000 g, resuspended in 400 µl of lysis buffer (200 mM NaCl, 50 mM KPi pH 7.5, 2 mM $MgCl_2$, 40 µg/mL DNAseI, 1 µM leupeptin, 1 µM pepstatin, 0.3 µM aprotinin, 1 mM benzamidine, 1 mM PMSF). Cells were lysed by disruption using sharp 500 µm glass beads. The supernatant was supplemented with DDM at a final concentration of 1% (w/v) and incubated for 1 hr at 4 °C under gentle agitation. The whole-cell extract was clarified by centrifugation at 200,000 g for 15 min and expressed proteins not containing a fusion to a fluorescent protein were detected by western blots. Constructs fused to a fluorescent protein were detected by in-gel fluorescence imaging (*Geertsma et al., 2008a*) and fluorescence size exclusion chromatography (FSEC) (*Kawate and Gouaux, 2006*) using a Superdex S200 column connected to an Agilent HPLC system (GE Healthcare) ($\lambda_{ex}$=489 nm / $\lambda_{em}$=510 nm). The NRMT homologue (WP_114552427.1) from the prokaryote *Eggerthella lenta* (DSM 2243, EleNRMT) was selected as the biochemically best-behaved homologue based on the profile of the eluted peak. The stability of EleNRMT was further improved by mutagenesis. To this end, consensus amino acids were identified using JALVIEW (*Waterhouse et al., 2009*) from aligned sequences of representative NRMT homologs, using Muscle (*Edgar, 2004*), and reported criteria (*Cirri et al., 2018*).

## Expression and purification of EleNRMT

For large-scale overexpression, the DNA coding for the NRMT from the prokaryote *Eggerthella lenta* (EleNRMT) was cloned into the pBXNH3 vector (Invitrogen) using the FX cloning technique (*Geertsma and Dutzler, 2011*) providing a construct where *EleNRMT* contains a 10-His tag and a HRV 3 C protease cleavage site at its N-terminus. All mutants were generated using the Quickchange site-direct mutagenesis method (Agilent). The vector was transformed into *E. coli* MC1061 cells and a preculture was grown in TB-Gly medium overnight using ampicillin (100 µg/ml) as a selection marker for all expression cultures. The preculture was diluted at 1:100 volume ratio into TB-Gly medium. The cells were grown at 37 °C to an $OD_{600}$ of 0.7–0.9, before the temperature was lowered to 25 °C. The expression was induced by addition of arabinose to a final concentration of 0.0045% (w/v) for 12–14 hr at 18 °C. Cells were harvested by centrifugation for 20 min at 5,000 g, resuspended in buffer RES and disrupted using a high-pressure cell lyser (Maximator HPL6). After low spin centrifugation (10,000 g for 20 min) to remove cell debris, the supernatant was subjected to an ultracentrifugation step (200,000 g for 45 min) and the pelleted membrane was resuspended in buffer EXT at a concentration of 0.5 g of vesicles per ml of buffer. The purification of EleNRMT and its mutants was carried out at 4 °C. Suspended membrane vesicles were diluted 1:2 in buffer EXT supplemented with protease inhibitors (Roche cOmplete EDTA-free) and 1% (w/v) detergent, depending on the purpose of the experiments. DDM (Anatrace) was used as detergent for proteoliposome-reconstitution and cryoEM sample preparation and DM (Anatrace) for ITC and crystallization. The extraction was carried out under gentle agitation for 2 hr at 4 °C. The lysate was cleared by centrifugation at 200,000 g for 30 min. The supernatant supplemented with 15 mM of imidazole at pH 7.5 was loaded onto NiNTA resin and incubated for at least 1 hr under gentle agitation. During later stages of purification, the detergent concentration was reduced to 0.04% (w/v) for DDM or 0.25% (w/v) for DM. The resin was washed with 20 CV of buffer W and protein was eluted with buffer ELU, both containing the respective detergent. The tag was cleaved by addition of HRV 3 C protease at a 3:1 molar ratio and dialyzed against buffer DIA containing the respective detergent for at least 2 hr at 4 °C. After addition of NiNTA resin to remove the cleaved tag, the sample was concentrated by centrifugation using a 50 kDa molecular weight cut-off concentrator (Amicon) and further purified by size exclusion chromatography on a Superdex S200 column (GE Healthcare) pre-equilibrated with buffer SEC containing the respective detergent. The peak fractions were pooled, concentrated, and used immediately for further experiments.

## Thermal stability assay using fluorescence-detection size-exclusion chromatography

The assay was performed following a published protocol (*Hattori et al., 2012*). Aliquots of 50 µl at 1 µM of EleNRMT and EleNRMT$^{ts}$ in 200 mM NaCl, 20 mM HEPES pH7, 0.04% DDM were incubated at temperatures up to 75 °C for 10 min. Aggregated protein was removed by centrifugal filtration (0.22 µm) and 25 µl of the sample was subsequently loaded onto a Superdex S200 column (GE Healthcare) equilibrated in 200 mM NaCl, 20 mM HEPES pH7, 0.04% DDM and coupled to a fluorescence detector (Agilent technologies 1200 series, G1321A). The proteins were detected using tryptophan fluorescence ($\lambda_{ex}$=280 nm; $\lambda_{em}$=315 nm) and stability was assessed by measuring the peak height of the monomeric peaks. The peak heights were normalized to the corresponding value from samples incubated at 4 °C (100% stability) and melting temperatures ($T_m$) were determined by fitting the curves to a sigmoidal dose-response equation.

## Reconstitution of EleNRMT and EcoDMT into proteoliposomes

EleNRMT, EcoDMT and all of their mutants were reconstituted into detergent-destabilized liposomes (*Geertsma et al., 2008b*). The lipid mixture made of POPE, POPG (Avanti Polar Lipids) at a weight ratio of 3:1: was washed with diethylether and dried under a nitrogen stream followed by exsiccation overnight. The dried lipids were resuspended in 100 mM KCl and 20 mM HEPES at pH7. After three freeze-thaw cycles, the lipids were extruded through a 400 nm polycarbonate filter (Avestin, LiposoFast-BAsic) and aliquoted into samples with a concentration of 45 mg/ml. The extruded lipids were destabilized by adding Triton X-100 and the protein was reconstituted at a lipid to protein ratio (w/w) of 50. After several incubation rounds with biobeads SM-2 (Biorad), the proteoliposomes were harvested the next day, resuspended in 100 mM KCl and 20 mM HEPES pH7 and stored at –80 °C.

## Isothermal titration calorimetry

A MicroCal ITC200 system (GE Healthcare) was used for Isothermal titration calorimetry experiments. All titrations were performed at 6 °C in buffer SEC (200 mM NaCl, 20 mM HEPES pH7, 0.25% DM). The syringe was filled with 3 mM MnCl$_2$ or MgCl$_2$ and the titration was initiated by the sequential injection of 2 µl aliquots into the cell filled with EleNRMT or mutants at a concentration of 70 µM. Data were analyzed with the Origin ITC analysis package and fitted assuming a single binding site for metal ions and a stoichiometry of N = 1. Each experiment was repeated twice from independent protein purifications with similar results. Experiments with buffer not containing any protein were performed as controls.

## Fluorescence-based substrate transport assays

Proteoliposomes were incubated with buffer IN (100 mM KCl, 20 mM HEPES pH 7) containing either 250 µM Calcein (Invitrogen), 100 µM Fura-2 (ThermoFischer Scientific), or 400 µM MagGreen (ThermoFischer Scientific) depending on the assayed ion. After three cycles of freeze-thawing, the liposomes were extruded through a 400 nm filter and the proteoliposomes were centrifuged and washed three times with buffer IN not containing any fluorophores. The proteoliposomes at a concentration of 25 mg/ml were subsequently diluted 1:100 to a final concentration of 0.25 mg/ml in buffer OUT (100 mM NaCl, 20 mM HEPES pH7) and aliquoted in 100 µl batches into a 96-well plate (ThermoFischer Scientific) and the fluorescence was measured every 4 s using a Tecan Infinite M1000 fluorimeter. A negative membrane potential of –118 mV was established after addition of valinomycin to a final concentration of 100 nM as a consequence of the 100-fold outwardly directed K$^+$ gradient. After 20 cycles of incubation, the substrate was added at different concentrations. The transport reaction was terminated by addition of calcimycin or ionomycin at a final concentration of 100 nM. The initial rate of transport was calculated by linear regression within the initial 100 s of transport after addition of the substrate. The calculated values were fitted to a Michaelis-Menten equation. Fura-2 was used to assay the transport of Ca$^{2+}$. For detection, the ratio between calcium bound Fura-2 ($\lambda_{ex}$=340 nm; $\lambda_{em}$=510 nm) and unbound Fura-2 ($\lambda_{ex}$=380 nm; $\lambda_{em}$=510 nm) was monitored during kinetic measurements. Calcein was used to assay the transport of Mn$^{2+}$ ($\lambda_{ex}$=492 nm; $\lambda_{em}$=518 nm) and Magnesium Green was to assay the transport of Mg$^{2+}$ ($\lambda_{ex}$=506 nm; $\lambda_{em}$=531 nm).

For assaying coupled proton transport, a proteoliposome stock of 15 mg/ml in 100 mM KCl, 5 mM HEPES pH 7, 50 µM ACMA was prepared. After clarification by sonication, the proteoliposomes were

diluted 1–100 to a concentration of 0.15 mg/ml in 100 mM NaCl, 5 mM HEPES pH7 and aliquoted into batches of 100 µl into a flat black 96-well plate (ThermoFischer Scientific). The fluorescence was measured every 4 s using a fluorimeter Tecan Infinite M1000 ($\lambda_{ex}$=412 nm; $\lambda_{em}$=482 nm). The transport reaction was initiated after addition of the substrate and valinomycin at a final concentration of 100 nM resulting in the same negative membrane potential of –118 mV as applied in assays that monitor transport of the metal ions. The reaction was terminated by addition of CCCP at 100 nM.

## Generation of nanobodies specific to EleNRMT

EleNRMT specific nanobodies were generated using a method described previously (*Pardon et al., 2014*). Briefly, an alpaca was immunized four times with 200 µg of detergent solubilized EleNRMT[ts]. Two days after the final injection, the peripheral blood lymphocytes were isolated and the total RNA fraction was extracted and converted into cDNA by reverse transcription. The nanobody library was amplified and cloned into the phage display pDX vector (Invitrogen). After two rounds of phage display, ELISA assays were performed on the periplasmic extract of 198 individual clones. Three nanobodies were identified. For all steps of phage display and ELISA, an enzymatically biotinylated version of EleNRMT with an additional Avi tag on its N-terminus (Avi-10His-3C) was used (*Fairhead and Howarth, 2015*) and bound to neutravidin coated plates. The biotinylation was confirmed by total mass spectrometry analysis.

## Expression and purification of nanobodies and preparation of the EleNRMT[ts]-Nb1,2 complex

All nanobodies were cloned into the pBXNPHM3 vector (Invitrogen) using the FX cloning technique. The expression construct contains the nanobody with a fusion of a pelB sequence, a 10-His tag, a maltose-binding protein (MBP) and a HRV 3 C protease site to its N-terminus. The vector was transformed into *E. coli* MC1061 cells and a preculture was grown in TB-Gly medium overnight using ampicillin (100 µg/ml) as a selection marker in all expression cultures. The preculture was used for inoculation of TB-Gly medium at 1:100 volume ratio. Cells were grown to an $OD_{600}$ of 0.7–0.9 and the expression was induced by addition of arabinose to a final concentration of 0.02% (w/v) for 4 hr at 37 °C. Cells were harvested by centrifugation for 20 min at 5000 g, resuspended in buffer A (200 mM KCl, 50 mM KPi pH 7.5, 10% Glycerol, 15 mM imidazole pH 7.5, 2 mM $MgCl_2$, 1 µM leupeptin, 1 µM pepstatin, 0.3 µM aprotinin, 1 mM benzamidine, 1 mM PMSF) and lysed using a high pressure cell lyser (Maximator HPL6). The purification of all nanobodies was carried out at 4 °C. The lysate was cleared by centrifugation at 200,000 g for 30 min. The supernatant was loaded onto NiNTA resin and incubated for at least 1 hr under gentle agitation. The resin was washed with 20 CV of buffer B (200 mM KCl, 20 mM Tris-HCl pH 8, 10% Glycerol, 50 mM imidazole, pH 7.5). The protein was eluted with buffer C (200 mM KCl, 20 mM Tris-HCl pH 8, 10% Glycerol, 300 mM imidazole, pH 7.5). The tag was cleaved by addition of HRV 3 C protease at a 3:1 molar ratio and dialyzed against buffer D (200 mM KCl, 20 mM Tris-HCl pH 8, 10% Glycerol) for at least 2 hr at 4 °C. After application of NiNTA resin to remove the tag, the sample was concentrated by centrifugation using a 10 kDa molecular weight cut-off concentrator (Amicon) and further purified by size exclusion chromatography using a Superdex S200 column (GE Healthcare) pre-equilibrated with buffer E (200 mM KCl, 20 mM Tris-HCl pH 8). The peak fractions were pooled, concentrated and flash-frozen and stored at –20 °C for subsequent experiments.

Following the purification of EleNRMT in DDM (for cryoEM data collection) or DM (for crystallization) as described, the purified membrane protein and nanobodies Nb1 and Nb2 were incubated at a 1:1.3:1.3 molar ratio for 30 min and the complex was further purified by size exclusion chromatography on a Superdex S200 column (GE Healthcare), pre-equilibrated in the same SEC buffer as used for EleNRMT purification. The sample was subsequently concentrated to 3.5 mg/ml (for cryoEM) or 8–12 mg/ml (for crystallization).

## Cryo-EM sample preparation and data collection

For sample preparation for cryo-EM, 2.5 µl of the complex were applied to glow-discharged holey carbon grids (Quantifoil R1.2/1.3 Au 200 mesh). Samples were blotted for 2–4 s at 4 °C and 100% humidity. The grids were frozen in liquid propane-ethane mix using a Vitrobot Mark IV (Thermo Fisher Scientific). For the dataset in presence of $Mg^{2+}$, the sample was supplemented with 10 mM $MgCl_2$ and incubated on ice for 20 min before grid preparation. All datasets were collected on a 300 kV

Titan Krios (ThermoFischer Scientific) with a 100 µm objective aperture and using a post-column BioQuantum energy filter with a 20 eV slit and a K3 direct electron detector in super-resolution mode. All datasets were recorded automatically using EPU2.9 with a defocus ranging from –1–2.4 µm, a magnification of 130,000 x corresponding to a pixel size of 0.651 Å per pixel (0.3255 Å in super resolution mode) and an exposure of 1.01 s (36 frames). Three datasets were collected for EleNRMT-Nb1,2 without $Mg^{2+}$ with respective total doses of 69.725, 61 and 69.381 $e^-/Å^2$ and for EleNRMT-Nb1,2 with $Mg^{2+}$ one dataset with a total dose of 69.554 $e^-/Å^2$.

## Cryo-EM data processing

Datasets were processed using Cryosparc v3.2.0 (*Punjani et al., 2017*) following the same processing pipeline (*Figure 5—figure supplements 1 and 2*). All movies were subjected to motion correction using patch motion correction with a fourier crop factor of 2 (pixel size of 0.651 Å/pix). After patch CTF estimation, high quality micrographs were identified based on relative ice thickness, CTF resolution and total full frame motion and micrographs not meeting the specified criteria were rejected. For the EleNRMT$^{ts}$-Nb1,2 complex, a partial dataset (981 micrographs) was used to generate a low-resolution 3D *ab initio* model. For this purpose, particles were selected using a blob picker with a minimum particle diameter of 160 Å and a minimum inter-particle distance of 64 Å. The selected particles were extracted with a box size of 360 pixels and binned 4 x (pixel size of 2.64 Å per pixel). These particles were initially used to generate 2D classes and later for template-driven particle picking. After extraction and 2D classification, an initial 3D ab initio model was generated in Cryosparc. This model was subsequently used to generate 2D templates for particle picking on the remainder of the dataset.

The 2D classes generated from particles selected from the entire dataset were extracted using a box size of 360 pix and down-sampled to 180 pixels (pixel size of 1.32 Å/pix). These particles were used for generation of 4 *ab initio* classes. Promising *ab initio* models were selected based on visual inspection and subjected to heterogenous refinement using one of the selected models as 'template' and an obviously bad model as decoy model. After several rounds of heterogenous refinement, the selected particles and models were subjected to non-uniform refinement (input model filtering to 8 Å) followed by local CTF refinement and another round of non-uniform refinement. Finally, the maps were sharpened using the sharpening tool from the Cryosparc package. The quality of the map was evaluated validated using 3DFSC (*Tan et al., 2017*) for FSC validation and local resolution estimation.

## Cryo-EM model building and refinement

Model building was performed in Coot (*Emsley and Cowtan, 2004*). Initially, the structure of ScaDMT (PDB 5M94) was rigidly fitted into the densities and served as template for map interpretation. The quality of the map allowed for the unambiguous assignment of residues 43–438. The structure of Nb16 (PDB 5M94) was used to build Nb1 and Nb2. The model was iteratively improved by real space refinement in PHENIX (*Afonine et al., 2018*) maintaining secondary structure constrains throughout. Figures were generated using ChimeraX (*Pettersen et al., 2021*) and Dino (http://www.dino3d.org). Surfaces were generated with MSMS (*Sanner et al., 1996*).

## Crystallization and X-ray structure determination of the EleNRMT$^{ts}$-Nb1,2 complex

The EleNRMT$^{ts}$-Nb1,2 complex was purified in DM as described before, supplemented with *E. coli* polar lipids solubilized in DM to a final concentration of 100 µg/ml and used for the preparation of vapor diffusion crystallization experiments in sitting drops by mixing 150 nl of the concentrated protein with 150 nl of the mother liquor containing 50 mM MgAc, 50 mM HEPES pH 7.2–7.6 and 25–30% PEG400. Crystals grew after 4 days and reached a maximum size of 0.2 × 0.3 x 0.3 mm after 12 days. The crystals were fished, cryoprotected by soaking into mother liquor supplemented with PEG400 to a final concentration of 35% and flash frozen into liquid nitrogen.

All data sets were collected on frozen crystals on either the X06SA or the X06DA beamline at the Swiss Light Source of the Paul Scherrer Institute (SLS, Villingen) on an EIGER 16 M or PILATUS 2 M-F detector (Dectris), respectively. Anomalous data were collected at the absorption edge of manganese (1.896 Å). Integration and scaling was performed using XDS (*Kabsch, 1993*) and data were further processed with CCP4 programs (*CCP4, 1994*). Phases were obtained by molecular replacement in Phaser (*McCoy et al., 2007*). For the ternary complex EleNRMT- Nb1-Nb2, the refined cryoEM

structure was used as search model for the respective datasets. Structures were refined in PHENIX (*Liebschner et al., 2019*) maintaining tight coordinate restraints and models were inspected and modified in Coot (*Emsley and Cowtan, 2004*).

## Acknowledgements

This research was supported by the Swiss National Science Foundation (SNF) through the National Centre of Competence in Research TransCure. We thank Dr. Nicolas Reyes for his help with the identification of positions for thermostabilization of EleNRMT and Dr. Marta Sawicka for input in cryo-EM and help during initial sample characterization. Nanobodies were generated with the help of the Nanobody Service Facility of UZH with the help of Dr. Sasa Stefanic. The assistance of Yvonne Neldner during library generation is acknowledged. The cryo-electron microscope and K3-camera were acquired with support of the Baugarten and Schwyzer-Winiker foundations and a Requip grant of the Swiss National Science Foundation. We thank Simona Sorrentino and the Center for Microscopy and Image Analysis (ZMB) of the University of Zurich for their support and access to the electron microscope and Beat Blattmann of the Protein Crystallization Center of the Department of Biochemistry for help with crystallization screening. X-ray data were collected at the X06SA and X06DA Beamlines at the Swiss Light Source of the Paul Scherrer Institute. All members of the Dutzler lab are acknowledged for help in various stages of the project.

## Additional information

### Funding

| Funder | Grant reference number | Author |
| --- | --- | --- |
| Swiss National Science Foundation | NCCR TransCure | Raimund Dutzler |

The funders had no role in study design, data collection and interpretation, or the decision to submit the work for publication.

### Author contributions

Karthik Ramanadane, Cloned, expressed and purified proteins, performed transport assays, prepared samples for X-ray crystallography and cryo-EM, processed cryo-EM data and built models, collected and processed ITC and X-ray data., Collected and processed ITC and X-ray data., Conceptualization, Data curation, Formal analysis, Investigation, Methodology, Validation, Visualization, Writing - original draft, Writing - review and editing; Monique S Straub, Collected cryo-EM data., Investigation, Methodology, Writing - review and editing; Raimund Dutzler, Conceptualization, Funding acquisition, Project administration, Supervision, Visualization, Writing - original draft, Writing - review and editing; Cristina Manatschal, Collected and processed ITC and X-ray data., Conceptualization, Data curation, Formal analysis, Investigation, Methodology, Project administration, Supervision, Validation, Visualization, Writing - original draft, Writing - review and editing

### Author ORCIDs

Karthik Ramanadane ⓘ http://orcid.org/0000-0001-7188-0250
Monique S Straub ⓘ http://orcid.org/0000-0002-7721-5048
Raimund Dutzler ⓘ http://orcid.org/0000-0002-2193-6129
Cristina Manatschal ⓘ http://orcid.org/0000-0002-4907-7303

### Decision letter and Author response

Decision letter https://doi.org/10.7554/eLife.74589.sa1
Author response https://doi.org/10.7554/eLife.74589.sa2

# Additional files

## Supplementary files
- Transparent reporting form
- Source data 1. Sequences, transport assays, ITC data, chromatograms, gels.

## Data availability

The cryo-EM density maps of the EleNRM-Nb complex in absence and presence of $Mg^{2+}$ have been deposited in the Electron Microscopy Data Bank under ID codes EMD-13985 and EMD-13987, respectively. The coordinates for the atomic model of the EleNRM-Nb complex in absence of $Mg^{2+}$ refined against the 3.4 Å cryo-EM density and the coordinates of the EleNRMTts-Nb1,2 complex in presence of $Mg^{2+}$ refined against the 4.1 Å cryo-EM density have been deposited in the Protein Data Bank under ID codes 7QIA and 7QIC. The coordinates and structure factors of the EleNRMTts-Nb1,2 complexes in $Mg^{2+}$ and $Mn^{2+}$ have been deposited in the Protein Data Bank with the accession codes 7QJI and 7QJJ. Source data files have been provided for Figures 1, Figure 1-figure supplement 1, Figure 2, Figure 2-figure supplement 1, Figure 3, Figure 2-figure supplement 2, Figure 4, Figure 4-figure supplement 1, Figure 8.

The following dataset was generated:

| Author(s) | Year | Dataset title | Dataset URL | Database and Identifier |
|---|---|---|---|---|
| Ramanadane K, Straub MS, Dutzler R, Manatschal C | 2021 | Structure of apo-EleNRMT in complex with two nanobodies at 3.5A | https://doi.org/10.2210/pdb7QIA/pdb | Worldwide Protein Data Bank, 10.2210/pdb7QIA/pdb |
| Ramanadane K, Straub MS, Dutzler R, Manatschal C | 2021 | Cryo-EM map of apo-EleNRMT in complex with two nanobodies at 3.5A | https://www.emdataresource.org/EMD-13985 | EMDataResource, 13985 |
| Ramanadane K, Straub MS, Dutzler R, Manatschal C | 2021 | Structure of magnesium-bound EleNRMT in complex with two nanobodies at 4.1A | https://doi.org/10.2210/pdb7QIC/pdb | Worldwide Protein Data Bank, 10.2210/pdb7QIC/pdb |
| Ramanadane K, Straub MS, Dutzler R, Manatschal C | 2021 | Cryo-EM map of magnesium-bound EleNRMT in complex with two nanobodies at 4.1A | https://www.emdataresource.org/EMD-13987 | EMDataResource, 13987 |
| Ramanadane K, Straub MS, Dutzler R, Manatschal C | 2022 | X-Ray Structure of apo-EleNRMT in complex with two Nanobodies at 4.1A | https://doi.org/10.2210/pdb7QJI/pdb | Worldwide Protein Data Bank, 10.2210/pdb7QJI/pdb |
| Ramanadane K, Straub MS, Dutzler R, Manatschal C | 2022 | X-Ray Structure of a Mn2+ soak of EleNRMT in complex with two Nanobodies at 4.6A | https://doi.org/10.2210/pdb7QJJ/pdb | Worldwide Protein Data Bank, 10.2210/pdb7QJJ/pdb |

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
