## [Editor Report]

This work elegantly fuses cryo-EM, X-ray crystallography, and in vitro transport experiments to describe the structural basis for functional diversity in the SLC11/NRAMP family of membrane transporters. This work identifies factors responsible for selectivity of classical NRAMPS for transition metal ions (Fe, Mn) and the NRMT clade for alkali metal ion (Mg). Although selectivity is much discussed in transport of divalent metal ions, this is an outstanding example of a study that gets to the bottom of the structural determinants governing this behavior.

---

## [Decision Letter]

**Decision letter after peer review:**

Thank you for submitting your article "Structural and functional properties of a magnesium transporter of the SLC11/NRAMP family" for consideration by *eLife*. Your article has been reviewed by 3 peer reviewers, including Randy B Stockbridge as Reviewing Editor and Reviewer #1, and the evaluation has been overseen by Kenton Swartz as the Senior Editor. The following individual involved in review of your submission has agreed to reveal their identity: Motoyuki Hattori (Reviewer #2).

Essential revisions:

1) A valinomycin/K^+^ gradient was used to establish a membrane potential for the transport assays. The exact magnitude of the K^+^ gradient in each assay is not clearly stated, but it appears that the K^+^ gradient may differ between the proton- and metal-uptake assays. Since membrane potential is known to influence transport by the NRAMPs, the manuscript should be revised to include the reasoning for applying a membrane potential in the main text, and to clearly describe the K^+^ gradients used in the methods. In addition, an extra control experiment should be performed to show that manganese/magnesium uptake occurs when the external K^+^ concentration is matched to that used for the proton uptake assays.

2) Since the applied membrane potential might drive non-specific substrate leak, no-protein controls should be shown for the highest concentrations of each substrate.

3) The reviewers are not fully convinced that the data support the conclusion that substitution of Trp for His is the sole determinant for proton coupling. It may be of interest to test a mutation at this site. However, if the authors choose not to do this, a more robust discussion of the proton transport pathway and a more tentative conclusion about the His to Trp change is warranted.

4) The reviewers had several suggestions for clarity of data presentation (described in the "Recommendations for authors"). These should be addressed.

*Reviewer #1 (Recommendations for the authors):*

1) A valinomycin/K^+^ gradient was used to establish a membrane potential in the transport assays. This aspect of the assay was only mentioned in the methods, and should be mentioned/discussed in the main text as well. Was this maneuver necessary for assay sensitivity (to drive accumulation of higher substrate concentrations for detection), or was it necessary for transport (to drive substrate translocation over an energy barrier?).

From the methods, it appears that the external K^+^ concentration in the uptake experiments is determined by the dilution of the proteoliposomes into the assay buffer, and liposomes were diluted to different extents for metal and proton uptake assays, ~6-fold difference. Assuming I understand the various dilutions correctly, the applied membrane potential driving force is likely reduced by ~50 mV in the proton-detection assays compared to the metal-detection assays. It has been shown that some NRAMPs have a fairly strong dependence on the membrane potential, such that a 50 mV change could impact the observation of transport (Bozzi et al., 2019, JGP 151: 1413-1429.) This loose end should be tied up by simply showing that manganese/magnesium uptake occurs when the external K^+^ concentration is matched to that used for the proton uptake assays.

2) Divalent metals are occasionally known to leak nonspecifically across liposomal membranes, and this would be enhanced under the conditions of these metal transport assays, in which a K^+^ gradient/valinomycin is used to drive uptake of the divalent metal ions into the liposomes. Therefore, it is important to show a no-protein control, at least at the highest concentrations of permeant substrate. Such controls were only shown for the proton uptake assays (which were negative results anyway).

3) At least in my rendering of the figures, for panels 2D, 4D, 4F, the datapoints for some datasets can't be seen because they are behind the shaded error envelope for another dataset. Even though the point is taken that these datasets are similar to each other, it's important to see the datapoints. Could this be visualized in some other way?

*Reviewer #2 (Recommendations for the authors):*

1. I understand that Trp229 in EleNRMT may account for proton-independent metal transport by EleNRMT. In this case, how does the transport cycle including the transition from the inward-facing to the outward-facing occur? Do the authors have any idea or speculation?

2. Page 5. "The uptake of Mg^2+^ into cells is thus accomplished by few selective transport systems (Payandeh et al., 2013) that include the bacterial proteins CorA (Lunin et al., 2006), CorC (Huang et al., 2021), MgtE (Tanaka et al., 2007) and their eukaryotic homologues (Schaffers et al., 2018; Schweigel-Rontgen and Kolisek, 2014), the TRP channel TRPM7 (Huang et al., 2020b) and Mg^2+^-selective P-type pumps (Maguire, 1992)."

The citation on MgtE (Tanaka et al., 2007) is a crystallization paper. I highly recommend the authors to correct the citation to the article on the MgtE structure (PMID: 17700703).

3. Page 16. "Finally, by replacing Gly 88 to asparagine in the background of the T224A mutant, we create an EleNRMT construct with a similar binding site signature as the EcoDMT mutant M235A, which showed increased ca^2+^ permeability."

This is a typo. It should be Gly58, not Gly 88.

4. Page 16. "In line with the pronounced effect already observed for T224A (Figure 8D, E), we see a further reduction of Mn^2+^ transport in the double mutant T224A/G88N (Figure 8H)."

This is a typo. It should be G58N, not G88N.

5. This is not a specific recommendation or request. Do the nobodies affect the transport activity of EleNRMT? This is just out of curiosity.

*Reviewer #3 (Recommendations for the authors):*

Despite my overall enthusiasm for the rigor of the work, there are a number of improvements that could be made in the presentation of data.

Although the arguments about changes at the ion binding site are compelling and logical, the depiction of densities and atomic coordination at this site are somewhat ineffective. The ion site is only shown in detail in Figure 5 Suppl 5 D and E. These figures compare apo and Mg-bound structures, but their differing orientations makes comparison difficult. The site is quite faded in both panels and it is not very easy to evaluate the presence/absence of Mg-associated density. This legend is misleading in referring to "X-ray structures of EleNRMT" whereas the figure actually illustrates both X-ray and cryo-EM structures.

In Figure 7, the authors try to illustrate a 1.5 A shift in the site that contributes to expansion of the ion binding site, but again this is ineffective. The molecular surface shown in the insets do not help me very much. Importantly, it is not clear which structure was used for this depiction. If the cryo-EM structure of the apo state has been used, which is likely given the higher resolution, then it is possible that the site has relaxed due to the absence of the ion, as can be been seen in apo structures of the NRAMPS.

The coordination of the hydrated Mg ion is finally depicted in Figure 9C, but again is difficult to appreciate from this rendering. I think that removal of the molecular surface and display of all the atomcs surrounding this site might make it easier to evaluate. It is not clear what panels E and F are trying to illustrate or the origin of the surface: is it based on model or map densities?

In Figure 6, the perspective and shadowing of the space-filling model in panel C could be improved. The authors should consider using a program like HOLLOW or HOLE or CAVER to produce an surface illustrating for the water-filled cavity. The text states that there is comparison of EleNRMT with both ScaDMT and DraNRAMP, but it appears that only ScaDMT is shown. Given the multiple structures of ScaDMT in the PDB, it would be very useful to identify the PDB code used for this figure. The color scheme is very muted and there is a strong fog on the perspective which makes lowers the visual impact of the comparison.

In Figure 7, the proton transport pathway is addressed in panels D and E by comparing structures of EleNRMT and ScaDMT. Although the functional data clearly show the lack of proton coupling in EleNRMT, this structural explanation not so convincing. It is not clear that the proposed proton pathway for NRAMPS is so specific that the authors conclusion about the replacement of His228 with a Trp is clear cut. Previous work from Bozzi et al., show effects of many site mutants on cell-based uptake, but no in vitro proton coupling assays. This issue seems complex and deserves further discussion and perhaps mutational data to support the critical importance of this His residue. Again, the PDB code for ScaDMT would be helpful.

Finally, for Figure 1, it would be nice to identify the various homologs that are discussed in the text to help the reader follow the discussion.

---

## [Author Response]

Essential revisions:1) A valinomycin/K^+^ gradient was used to establish a membrane potential for the transport assays. The exact magnitude of the K^+^ gradient in each assay is not clearly stated, but it appears that the K^+^ gradient may differ between the proton- and metal-uptake assays. Since membrane potential is known to influence transport by the NRAMPs, the manuscript should be revised to include the reasoning for applying a membrane potential in the main text, and to clearly describe the K^+^ gradients used in the methods. In addition, an extra control experiment should be performed to show that manganese/magnesium uptake occurs when the external K^+^ concentration is matched to that used for the proton uptake assays.

The negative membrane potential in our transport experiments was applied to increase the sensitivity of the assays as illustrated in Author response image 1 for Mn^2+^ transport mediated by EcoDMT. As shown, the application of a membrane potential of -118 mV, which was established in response to the addition of valinomycin to liposomes suspended in solutions establishing a 100-fold K^+^ gradient (with liposomes containing 100 mM KCl on their inside and 1 mM on their outside) enhances transport and increases the apparent K_m_. We use an equivalent protocol for all transport experiments shown in our manuscript irrespective of the monitored substrate. Hence, experiments assaying metal ion transport and those that monitor H^+^ transport were performed with the same membrane potential applied. In all cases, this was achieved by diluting a proteoliposome stock solution suspended in 100 mM KCl at a ratio of 1:100 into a solution containing 100 mM NaCl.

**Author response image 1. sa2fig1:** Voltage dependence of Mn^2+^ transport by EcoDMT. a, Mn^2+^ transport assayed at a membrane potential of 0 mV. b, Mn^2+^ transport assayed at a membrane potential of -118 mV. a, b Mn^2+^ uptake into proteoliposomes was monitored by the quenching of calcein trapped inside the liposomes. Traces measured with indicated Mn^2+^ concentrations added to the outside are shown in unique colors.

To better emphasize this point, we added the reasoning for applying a membrane potential in the Results section and clarified the protocol of liposome preparation for the assays in the methods section.

Line 134-142

“All transport experiments were carried out in in presence of a 100-fold outwardly directed K^+^ gradient, which after addition of the ionophore valinomycin establishes a membrane potential of -118 mV, to enhance the sensitivity of the applied assays.”

Line 668-674

“The proteoliposomes at a concentration of 25 mg/ml were subsequently diluted 1 to 100 to a final concentration of 0.25 mg/ml in buffer OUT (100 mM NaCl, 20 mM HEPES pH7) and aliquoted in 100 μl batches into a 96-well plate (ThermoFischer Scientific) and the fluorescence was measured every 4 seconds using a Tecan Infinite M1000 fluorimeter. A negative membrane potential of -118 mV was established after addition of valinomycin to a final concentration of 100 nM as a consequence of the 100-fold outwardly directed K^+^ gradient.”

2) Since the applied membrane potential might drive non-specific substrate leak, no-protein controls should be shown for the highest concentrations of each substrate.

There is no detectable non-specific substrate leak into liposomes not containing protein. We have added the respective no-protein controls for the highest concentrations of each substrate (400 μM Mn^2+^, 2 mM ca^2+^ and 5 mM Mg^2+^) as Figure 2—figure supplement 1.

3) The reviewers are not fully convinced that the data support the conclusion that substitution of Trp for His is the sole determinant for proton coupling. It may be of interest to test a mutation at this site. However, if the authors choose not to do this, a more robust discussion of the proton transport pathway and a more tentative conclusion about the His to Trp change is warranted.

Our transport assay demonstrated the absence of H^+^ transport in EleNRMT and we have correlated this observation to structural features that were previously proposed to be connected to H^+^ transport in pro- and eukaryotic SLC11 family members, which are all absent in EleNRMT. It was not our intention to speculate on the mechanism of H^+^ transport in classical H^+^-coupled family members as this is not the subject of our study. In no way we did want to imply that the single replacement of the mentioned tryptophane (TRP 229) located on α-helix 6 for a histidine would restore H^+^ transport, particularly since our study has shown that a change in selectivity can generally not be accomplished by a single point mutation in the protein.

To clarify this point we have reworded the paragraph in the results:

Line 311-322:

“Proton transport in classical NRAMP transporters was proposed to be linked to structural features extending from the ion binding site (Bozzi et al., 2019a; Bozzi et al., 2020; Bozzi et al., 2019b; Ehrnstorfer et al., 2017; Mackenzie et al., 2006; Pujol-Gimenez et al., 2017). These include the conserved aspartate involved in the coordination of metal ions, two conserved histidines on α-helix 6b downstream of the binding site methionine and a continuous path of acidic and basic residues on α-helices 3 and 9 surrounding a narrow aqueous cavity, which together appear to constitute an intracellular H^+^ release pathway. Apart from the binding site aspartate (Asp 55) and glutamate on α-helix 3 (Glu 133) close to the metal ion binding site, all other positions in EleNRMT are altered to residues that cannot accept protons. The two histidines on α-helix 6b are substituted by a tryptophane and an asparagine, respectively, and the polar pathway residues on α-helix 3 and 9 are replaced by hydrophobic residues and a threonine (Figure 7B, D. Figure 3—figure supplement 1B).”

4) The reviewers had several suggestions for clarity of data presentation (described in the "Recommendations for authors"). These should be addressed.

We have addressed all suggestions below and, where appropriate, introduced corresponding changes to our manuscript.

Reviewer #1 (Recommendations for the authors):1) A valinomycin/K^+^ gradient was used to establish a membrane potential in the transport assays. This aspect of the assay was only mentioned in the methods, and should be mentioned/discussed in the main text as well. Was this maneuver necessary for assay sensitivity (to drive accumulation of higher substrate concentrations for detection), or was it necessary for transport (to drive substrate translocation over an energy barrier?).

The established membrane potential was applied to increase assay sensitivity. This is now clarified in the results (see essential reviews point 1).

From the methods, it appears that the external K^+^ concentration in the uptake experiments is determined by the dilution of the proteoliposomes into the assay buffer, and liposomes were diluted to different extents for metal and proton uptake assays, ~6-fold difference. Assuming I understand the various dilutions correctly, the applied membrane potential driving force is likely reduced by ~50 mV in the proton-detection assays compared to the metal-detection assays. It has been shown that some NRAMPs have a fairly strong dependence on the membrane potential, such that a 50 mV change could impact the observation of transport (Bozzi et al., 2019, JGP 151: 1413-1429.) This loose end should be tied up by simply showing that manganese/magnesium uptake occurs when the external K^+^ concentration is matched to that used for the proton uptake assays.

The same membrane potential (-118 mV) was applied in all experiments. This is clarified in the results and the methods of our revision (see essential reviews point 1).

2) Divalent metals are occasionally known to leak nonspecifically across liposomal membranes, and this would be enhanced under the conditions of these metal transport assays, in which a K^+^ gradient/valinomycin is used to drive uptake of the divalent metal ions into the liposomes. Therefore, it is important to show a no-protein control, at least at the highest concentrations of permeant substrate. Such controls were only shown for the proton uptake assays (which were negative results anyway).

We are now showing the non-specific background for metal ion transport, which is in all cases negligible (see essential reviews point 2).

3) At least in my rendering of the figures, for panels 2D, 4D, 4F, the datapoints for some datasets can't be seen because they are behind the shaded error envelope for another dataset. Even though the point is taken that these datasets are similar to each other, it's important to see the datapoints. Could this be visualized in some other way?

We have tried to change this by moving the mean values to the front and the errors to the back. In that way we hope that we have allowed a better view of all measurements. However, some overlap is unavoidable in cases where different conditions do not affect transport. The indicated panels were replaced in our revised manuscript.

Reviewer #2 (Recommendations for the authors):1. I understand that Trp229 in EleNRMT may account for proton-independent metal transport by EleNRMT. In this case, how does the transport cycle including the transition from the inward-facing to the outward-facing occur? Do the authors have any idea or speculation?

It is common that representatives of transport proteins sharing the same general fold include secondary-active and uncoupled transporters. Examples for this behavior has been found for transporters of the major-facilitator superfamily, ones that share an UraA-like fold and also for proteins sharing a LeuT fold that include members of the SLC11/NRAMP family. In all cases, coupled and uncoupled transporters undergo equivalent conformational transitions, which in case of the SLC11 family are illustrated in different structures showing inward- occluded and outward-facing conformations. In case of an uncoupled transporter, the protein is able to sample these conformations irrespectively of whether its binding site is occupied. In contrast, in coupled transporters certain transitions are energetically unfavorable, such as that of a partially loaded transporter in symporters and of an unloaded transporter in case of exchangers.

In this respect it is noteworthy that a lose coupling of Fe^2+^ transport to protons has been proposed for the human transporter DMT1 based on electrophysiology (Mackenzie et al., 2006). In this case the transporter would also be able to change its conformations in a partially loaded state, which would be highly unfavorable in case of a strictly coupled symporter. As detailed above, we do not want to imply that the His replacing Trp229 in H^+^ coupled family members would be the only feature conferring H^+^ coupling as discussed in detail before. Homology models of EleNRMT in occluded and outward-facing conformations, where Trp 229 would be buried in the protein, do not show any indication of steric hindrance by this bulky sidechain.

2. Page 5. "The uptake of Mg^2+^ into cells is thus accomplished by few selective transport systems (Payandeh et al., 2013) that include the bacterial proteins CorA (Lunin et al., 2006), CorC (Huang et al., 2021), MgtE (Tanaka et al., 2007) and their eukaryotic homologues (Schaffers et al., 2018; Schweigel-Rontgen and Kolisek, 2014), the TRP channel TRPM7 (Huang et al., 2020b) and Mg^2+^-selective P-type pumps (Maguire, 1992)."The citation on MgtE (Tanaka et al., 2007) is a crystallization paper. I highly recommend the authors to correct the citation to the article on the MgtE structure (PMID: 17700703).

We thank the reviewer for the note. We changed the citation accordingly.

3. Page 16. "Finally, by replacing Gly 88 to asparagine in the background of the T224A mutant, we create an EleNRMT construct with a similar binding site signature as the EcoDMT mutant M235A, which showed increased ca^2+^ permeability."This is a typo. It should be Gly58, not Gly 88.

We have corrected the typo.

4. Page 16. "In line with the pronounced effect already observed for T224A (Figure 8D, E), we see a further reduction of Mn^2+^ transport in the double mutant T224A/G88N (Figure 8H)."This is a typo. It should be G58N, not G88N.

We have corrected the typo.

5. This is not a specific recommendation or request. Do the nobodies affect the transport activity of EleNRMT? This is just out of curiosity.

We have investigated the effect of one of the two nanobodies, Nb1, on Mn^2+^ transport of EleNRMT. When added to the outside of proteoliposomes, Nb1 indeed inhibits Mn^2+^ transport in a concentration-dependent manner. However, in this case transport is not fully inhibited and saturates at about half of the maximum transport activity (Author response image 2). Complete inhibition can be achieved by adding Nb1 to both sides of the proteoliposomes (Author response image 2). Knowing that Nb1 binds to the extracellular side of EleNRMT, these results reflect the about equal distribution of transporters in inside-out an outside-out configurations.

**Author response image 2. sa2fig2:** Inhibition of EleNRMT mediated Mg^2+^ transport by Nb1. a, Mn^2+^ transport into proteoliposomes containing EleNRMT upon addition of indicated concentrations of Nb1 to the outside. b, Mn^2+^ transport into proteoliposomes containing EleNRMT upon addition of indicated concentrations of Nb1 added to both sides of the liposome. Partial inhibition reflects the about equal distribution of orientation of EleNRMT within the liposomes.

Reviewer #3 (Recommendations for the authors):Despite my overall enthusiasm for the rigor of the work, there are a number of improvements that could be made in the presentation of data.Although the arguments about changes at the ion binding site are compelling and logical, the depiction of densities and atomic coordination at this site are somewhat ineffective. The ion site is only shown in detail in Figure 5 Suppl 5 D and E. These figures compare apo and Mg-bound structures, but their differing orientations makes comparison difficult. The site is quite faded in both panels and it is not very easy to evaluate the presence/absence of Mg-associated density. This legend is misleading in referring to "X-ray structures of EleNRMT" whereas the figure actually illustrates both X-ray and cryo-EM structures.

We want to emphasize that we do not have any evidence for changes in the protein structure as a consequence of ion binding, as judged from a comparison of the cryo-EM structures of EleNRMT with and without addition of Mg^2+^. Similarly, the X-ray structures obtained in Mg^2+^ and Mn^2+^ do not show evidence for an altered backbone structure, although the limited resolution in these cases prohibited a detailed interpretation of sidechain conformations. This behavior resembles the case of ScaDMT, where the inward-facing structure of the protein did not show pronounced changes in its conformation irrespective of the binding of Mn^2+^. In all cases, it should be emphasized that smaller changes in side-chain orientation would not be detected at the comparably low resolution of the data. In that respect, we consider the anomalous density of Mn^2+^ in the X-ray structure as the strongest proof for metal ion binding at the described site. This density is shown in Figure 5C. We wanted to report about residual density in the cryo-EM structures, which are found at similar location as the observed Mn^2+^ position, as we consider it of potential relevance for Mg^2+^ binding. However, it should be emphasized that the experimental datasets at this resolution do not permit to draw definite conclusions and we thus decided to show this density as panels in Figure 5—figure supplement 5D and E. In these panels we wanted to emphasize the location of residual density (indicated by red and light green spheres) compared to the Mn^2+^ position (displayed as black spheres) rather than the coordination by surrounding residues, which were displayed to show the general features of the map.

In our revised manuscript we have replaced the panels of Figure 5—figure supplement 5 D and E to show both densities in the same orientation and labeled specific residues. We also changed the title of the figure to:

X-ray structures of EleNRMT-nanobody complexes in presence of Mg^2+^ and Mn^2+^ and comparison to cryo-EM structures.

In Figure 7, the authors try to illustrate a 1.5 A shift in the site that contributes to expansion of the ion binding site, but again this is ineffective. The molecular surface shown in the insets do not help me very much. Importantly, it is not clear which structure was used for this depiction. If the cryo-EM structure of the apo state has been used, which is likely given the higher resolution, then it is possible that the site has relaxed due to the absence of the ion, as can be been seen in apo structures of the NRAMPS.

The difference in the backbone structure between ScaDMT and EleNRMT is difficult to visualize since it consists of small delocalized changes over the entire binding site. We have tried to better illustrate this in our revision of Figure 7A where we now show two orientations. We have also altered the coloring of the insets in Figure 7B, C. The point we want to make here is that the bound ion in EleNRMT is located within the aqueous cavity, whereas it is surrounded by interacting protein chains in ScaDMT and thus located outside of the wide cavity, which should be obvious from the figure panels. We have also added the following sentence to the legend:

Line 1281-1283:

“Whereas the ion in EleNRMT is located inside the aqueous cavity, its location in ScaDMT is outside of the cavity, tightly surrounded by coordinating residues.”

As mentioned above, there are no appreciable differences between both cryo-EM structures and no relaxation of the site has been observed. Similarly, we did not observe any appreciable differences at the resolution of the data between the structures of ScaDMT in presence and absence of Mn^2+^.

The coordination of the hydrated Mg ion is finally depicted in Figure 9C, but again is difficult to appreciate from this rendering. I think that removal of the molecular surface and display of all the atomcs surrounding this site might make it easier to evaluate. It is not clear what panels E and F are trying to illustrate or the origin of the surface: is it based on model or map densities?

We want to emphasize that Figure 9C is a discussion figure that shows a plausible model where the Mg^2+^ ion was placed at the position of Mn^2+^ identified in the anomalous difference map and the water atoms were modeled based on an ideal octahedral conformation of hydrated Mg^2+^. Panels E and F depict a hypothetical model of an occluded conformation of EleNRMT that was constructed based on the structure of the corresponding conformation of DraNRAMP. The point we want to make is that, also in an occluded state, the ion binding site seems large enough to accommodate a hydrated Mg^2+^ ion.

In our revised manuscript, we have removed the surface in Figure 9C and clarified that the pocket in Figure E and F originate from a hypothetical model.

In Figure 6, the perspective and shadowing of the space-filling model in panel C could be improved. The authors should consider using a program like HOLLOW or HOLE or CAVER to produce an surface illustrating for the water-filled cavity. The text states that there is comparison of EleNRMT with both ScaDMT and DraNRAMP, but it appears that only ScaDMT is shown. Given the multiple structures of ScaDMT in the PDB, it would be very useful to identify the PDB code used for this figure. The color scheme is very muted and there is a strong fog on the perspective which makes lowers the visual impact of the comparison.

We have replaced Figure 6C with ribbon representations of the molecule where only the water-filled cavity accessing the binding site is depicted as surface representation. The figure shows a comparison with ScaDMT and the corresponding PDB code was added to the legend. We have tried different presentations for the superposition shown in Figure 6B and think that the current illustration yielded the best impression of the structural similarity between displayed molecules. In our view a change in the coloring of ScaDMT and a decrease of the depth would make a distinction even more difficult. The panel should make the point that EleNRMT and ScaDMT both adopt and inward-facing conformation. It is not intended to show the detailed difference in each of the aligned helices.

In Figure 7, the proton transport pathway is addressed in panels D and E by comparing structures of EleNRMT and ScaDMT. Although the functional data clearly show the lack of proton coupling in EleNRMT, this structural explanation not so convincing. It is not clear that the proposed proton pathway for NRAMPS is so specific that the authors conclusion about the replacement of His228 with a Trp is clear cut. Previous work from Bozzi et al., show effects of many site mutants on cell-based uptake, but no in vitro proton coupling assays. This issue seems complex and deserves further discussion and perhaps mutational data to support the critical importance of this His residue. Again, the PDB code for ScaDMT would be helpful.

See detailed comment in our response to essential revisions point 3. Our work does not aim to address the H^+^ coupling mechanism of classical NRAMP transporters. Instead, we are comparing conserved residues in NRAMP transporters that were previously suggested to participate in H^+^ transport, to their corresponding positions in EleNRMT. All relevant studies are referred to in our manuscript. It is remarkable that nearly all of the referred residues are altered in EleNRMT. As also illustrated in panels 7D and E we never intended to reduce the capability of H^+^ transport to the presence of a single residue.

Finally, for Figure 1, it would be nice to identify the various homologs that are discussed in the text to help the reader follow the discussion.

In our revised manuscript, we have labeled the described homologs in the phylogenetic tree shown in Figure 1.

References

Bozzi, A.T., L.B. Bane, C.M. Zimanyi, and R. Gaudet. 2019a. Unique structural features in an Nramp metal transporter impart substrate-specific proton cotransport and a kinetic bias to favor import. J Gen Physiol. 151:1413-1429.

Bozzi, A.T., A.L. McCabe, B.C. Barnett, and R. Gaudet. 2020. Transmembrane helix 6b links proton and metal release pathways and drives conformational change in an Nramp-family transition metal transporter. J Biol Chem. 295:1212-1224.

Bozzi, A.T., C.M. Zimanyi, J.M. Nicoludis, B.K. Lee, C.H. Zhang, and R. Gaudet. 2019b. Structures in multiple conformations reveal distinct transition metal and proton pathways in an Nramp transporter. eLife. 8.

Ehrnstorfer, I.A., C. Manatschal, F.M. Arnold, J. Laederach, and R. Dutzler. 2017. Structural and mechanistic basis of proton-coupled metal ion transport in the SLC11/NRAMP family. Nat Commun. 8:14033.

Hattori, M., R.E. Hibbs, and E. Gouaux. 2012. A fluorescence-detection size-exclusion chromatography-based thermostability assay for membrane protein precrystallization screening. Structure. 20:1293-1299.

Mackenzie, B., M.L. Ujwal, M.H. Chang, M.F. Romero, and M.A. Hediger. 2006. Divalent metal-ion transporter DMT1 mediates both H+ -coupled Fe2+ transport and uncoupled fluxes. Pflugers Arch. 451:544-558.

Pujol-Gimenez, J., M.A. Hediger, and G. Gyimesi. 2017. A novel proton transfer mechanism in the SLC11 family of divalent metal ion transporters. Sci Rep. 7:6194.